# Surrogate Assisted Generation
# of Human-Robot Interaction Scenarios

**Varun Bhatt, Heramb Nemlekar, Matthew C. Fontaine, Bryon Tjanaka,**
**Hejia Zhang, Ya-Chuan Hsu, and Stefanos Nikolaidis**
University of Southern California
Los Angeles, CA
`vsbhatt,nemlekar,mfontain,tjanaka,hejiazha,yachuanh,nikolaid@usc.edu`

**Abstract:** As human-robot interaction (HRI) systems advance, so does the difficulty of evaluating and understanding the strengths and limitations of these systems in different environments and with different users. To this end, previous methods have algorithmically generated diverse scenarios that reveal system failures in a shared control teleoperation task. However, these methods require directly evaluating generated scenarios by simulating robot policies and human actions. The computational cost of these evaluations limits their applicability in more complex domains. Thus, we propose augmenting scenario generation systems with surrogate models that predict both human and robot behaviors. In the shared control teleoperation domain and a more complex shared workspace collaboration task, we show that surrogate assisted scenario generation efficiently synthesizes diverse datasets of challenging scenarios. We demonstrate that these failures are reproducible in real-world interactions.

**Keywords:** Scenario Generation, Human-Robot Interaction, Quality Diversity

## 1   Introduction

As the complexity of robotic systems that interact with people increases, it becomes impossible for designers and end-users to anticipate how a robot will act in different environments and with different users. For instance, consider a robotic arm collaborating with a user on a package labeling task, where a user attaches a label while the robot presses a stamp with the goal of completing the task as fast as possible (Fig. 1). In this task, the arm infers the user's intended goal object and moves simultaneously towards a different object to avoid collision. The robot's motion depends on which object the user selects to label, how the user moves towards that object, and how all objects are arranged in the environment. Thus, evaluating the system requires testing it with a diverse range of user behaviors and object arrangements.

While user studies are essential for understanding how users will interact with a robot, they are limited in the number of environments and user behaviors they can cover. Algorithmically generating scenarios with simulated robot and human behaviors in an "Oz of Wizard" paradigm [1] can complement user studies by finding failures and elucidating a holistic view of the strengths and limitations of a robotic system's behavior.

Previous work [2, 3] has formulated algorithmic scenario generation as a quality diversity (QD) problem and demonstrated the effectiveness of QD algorithms in generating diverse collections of scenarios in a shared control teleoperation domain. In that domain, a user teleoperates a robotic arm with a joystick interface, while the robot observes the joystick inputs to infer the user's goal and assist the user in reaching their goal. However, these interactions only last a few seconds in contrast to collaborative, sequential tasks that last much longer. For instance, completing the package labeling task (Fig. 1) can take several minutes, making their evaluation expensive. This limits the applicability of QD algorithms, which require a large number of evaluations [4].

7th Conference on Robot Learning (CoRL 2023), Atlanta, USA.

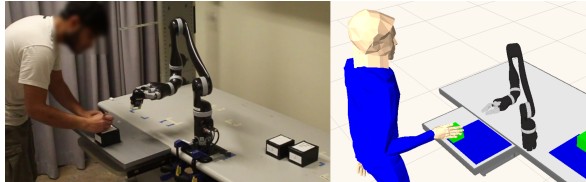

Figure 1: Example scenario in a collaborative package labeling task found by our proposed surrogate assisted scenario generation framework. The presence of the two objects behind the robot results in its expected cost-minimizing policy to move towards the object in the front, resulting in a conflict with the user who is reaching the object at the same time.

Our key insight is that *we can train deep neural networks as surrogate models to predict human-robot interaction outcomes and integrate them into the scenario generation process*. In addition to making scenario evaluations inexpensive, deep neural networks are end-to-end differentiable, which allows us to integrate state-of-the-art differentiable quality diversity (DQD) algorithms [5] to efficiently discover scenarios that the surrogate model predicts are challenging with diverse behavior.

We make the following contributions: (1) We introduce using deep neural networks as surrogate models to predict human-robot interaction outcomes, such as time to task completion, maximum robot path length, or total waiting time; (2) We integrate surrogate models with differentiable quality diversity (DQD) algorithms that leverage gradient information backpropagated through the surrogate model. (3) We show, in the shared control teleoperation domain of previous work [2] and in a shared workspace collaboration domain [6], that surrogate assisted scenario generation results in significant benefits in terms of sample efficiency. It also achieves a significant reduction in computational time in the collaboration domain, where evaluations are particularly expensive.

## 2 Problem Statement

We model the problem of generating a diverse and challenging dataset of human-robot interaction scenarios as a quality diversity (QD) problem and adopt the QD definition from prior work [5].

We assume a scenario parameterized by $\boldsymbol{\theta} \in \mathbb{R}^n$. The scenario parameters could be object positions and types in the environment, human model parameters, or latent inputs to a generative model of environments, which is converted to a scenario via a function $G(\boldsymbol{\theta})$. The objective function $f : \mathbb{R}^n \to \mathbb{R}$ assesses the quality of a scenario $\boldsymbol{\theta}$. Because we wish to find challenging scenarios, higher quality implies worse team performance, e.g., longer task completion time.

We further assume a set of user-defined measure functions, $m_i : \mathbb{R}^n \to \mathbb{R}$, or as a vector function $\boldsymbol{m} : \mathbb{R}^n \to \mathbb{R}^k$, that quantify aspects of the scenario that we wish to diversify, e.g., distance between objects, noise in user inputs, or human and robot path length. The range of $\boldsymbol{m}$ forms a measure space $S = \boldsymbol{m}(\mathbb{R}^n)$, which we assume is tessellated into $M$ cells, forming an *archive*.

The QD objective is to maximize the QD-score [7]: $\max_{\boldsymbol{\theta}_i} \sum_{i=1}^{M} f(\boldsymbol{\theta}_i)$. Here $\boldsymbol{\theta}_i$ refers to the scenario with the highest quality in cell $i$ of the archive. If there are no scenarios in a cell, $f(\boldsymbol{\theta}_i)$ is assumed to be zero.

The differentiable quality diversity (DQD) problem formulation is a special case of QD where the objective function $f$ and measure functions $\boldsymbol{m}$ are first-order differentiable.

## 3 Background

**Scenario Generation.** Algorithmic scenario generation has many applications, which include designing video game levels [8, 9, 10, 11, 12, 13, 14] and testing autonomous vehicles [15, 16, 17, 18, 19, 20, 21], motion planning algorithms [22], and reinforcement learning agents [23, 24, 25, 26]. It has also been applied to create curricula for robot learning [27, 28, 29, 30] and to co-evolve agents

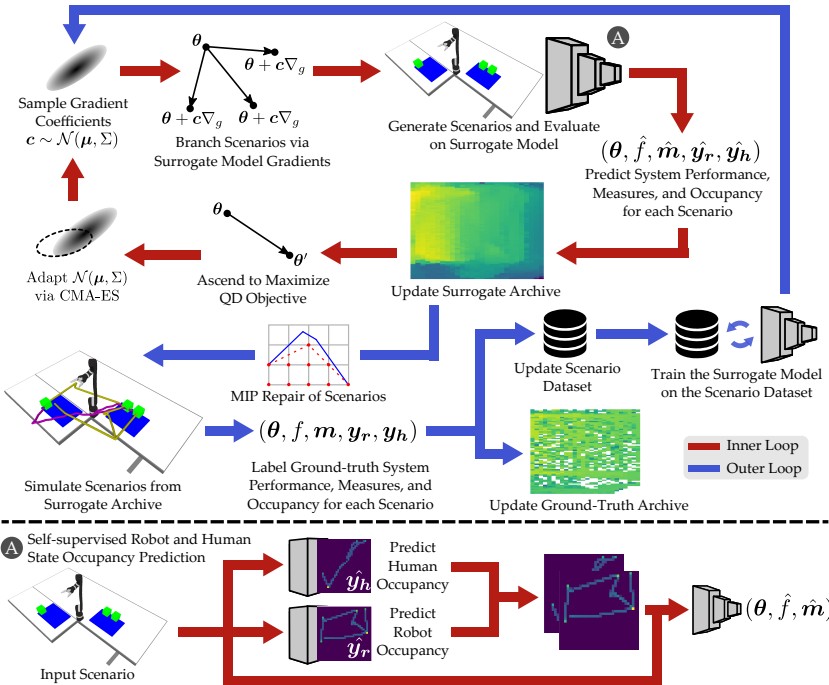

Figure 2: An overview of our proposed differentiable surrogate assisted scenario generation (DSAS) algorithm for HRI tasks. The algorithm runs in two stages: an inner loop to exploit a surrogate model of the human and the robot behavior (**red arrows**) and an outer loop to evaluate candidate scenarios and add them to a dataset (**blue arrows**). See App. A for the complete pseudocode.

and environments for agent generalizability [31, 32, 33, 34, 35, 36, 37, 38, 39, 40]. Most relevant to our work is prior work [2, 3] in human-robot interaction that applied the MAP-Elites [41] and CMA-ME [42] quality diversity (QD) algorithms to find robot failures in a shared control teleoperation domain. In this work, we significantly improve sample and wall-clock efficiency by combining the state-of-the-art QD algorithms CMA-MAE [43] and CMA-MAEGA [43] with surrogate models that predict human-robot interaction outcomes.

**QD Algorithms.** QD algorithms, such as MAP-Elites [41] and CMA-MAE [42, 43], solve the QD problem defined in Sec. 2. They have been used to generate diverse locomotion strategies [41, 44], video game levels [45], nano-materials [46], and building layouts [47]. Certain prior QD algorithms [4, 48, 49, 23] have leveraged surrogate models based on Gaussian processes or deep neural networks to guide the search. Most relevant to our work is the Deep Surrogate Assisted Generation of Environments (DSAGE) [23] algorithm, which exploits a surrogate model with quality diversity algorithms to generate environments. However, DSAGE has only been applied in *single-agent grid-world* game domains. In contrast, this work addresses the much more complex task of human-robot interaction scenarios, which requires the advances in the surrogate model, QD search, and scenario generation described in Sec. 4.

# 4 Surrogate Assisted Scenario Generation

Our method for algorithmically generating diverse collections of HRI scenarios builds upon recent work in generating single-agent grid-world environments with surrogate models [23]. We briefly describe the advances necessary to scale these techniques from single-agent grid-world game domains to the much more complex HRI domains.

**Surrogate Models for Human-Robot Interaction.** We scale the surrogate model to predict outcomes of HRI scenarios that include both robot and human behavior and environment parameters.

First, we allow both the environment and the human model parameters as inputs to the surrogate model. Second, we discretize the shared workspace and predict two occupancy grids, one for the human and one for the robot. We then stack both predictions as inputs to a convolutional neural network, which predicts the objective and measure functions. App. B provides precise details of our surrogate model.

**Scenario Repair via Mixed Integer Programming.** In contrast to previous work [2] that considered a single workspace, scenarios in our work have disjoint workspaces, with each workspace imposing constraints on object arrangement, such as boundary and collision avoidance constraints. Simulating the scenario is impossible if these constraints are not satisfied. We thus adopt a generate-then-repair approach [50, 51], where we generate unconstrained scenarios and pass them through a mixed integer program (MIP). We propose a MIP formulation that solves for the minimum cost edit of object locations – the sum of object displacement distances – that satisfies the constraints for a valid scenario. See App. C for the complete MIP formulation.

**Objective Regularization.** The generate-then-repair strategy repairs invalid scenarios by moving objects placed outside the workspace boundaries to the edge of the workspaces. However, this does not incentivize the QD algorithm to search the workspace interiors, which can result in the search diverging away from the workspace areas. To guide the search towards the workspace interiors, we discount the objective function $f$ of the QD formulation by the cost of the MIP repair. An ablation study in App. H.3 shows the effects of objective regularization on performance.

**DQD with Surrogate Models.** The DSAGE algorithm exploits the surrogate model with *derivative-free* QD algorithms. A key observation is that the surrogate model is an end-to-end differentiable neural network. We can take advantage of this by exploiting the surrogate model with differentiable quality diversity (DQD) algorithms [5], which leverage the gradients of the objective and measure functions to accelerate QD optimization. Leveraging DQD also lets us scale to higher dimensional scenario parameter spaces since the search is applied over the objective-measure space ($k + 1$ dimensions) instead of the scenario parameter space ($n$ dimensions).

**Algorithm.** Fig. 2 provides an overview of the complete algorithm, which consists of an outer loop (blue arrows) and an inner loop (red arrows). In the inner loop, a QD algorithm searches for scenarios that are challenging and diverse according to the surrogate model predictions. We repair the generated scenarios to ensure validity and evaluate each repaired scenario to obtain ground truth objective and measure values. Based on these values, we add each scenario to a ground-truth archive and a training dataset for the surrogate model. The surrogate model trains on this dataset, correcting prediction errors exploited by the QD algorithm. After accumulating enough diverse data, the surrogate model starts making accurate predictions, and the inner loop produces truly diverse and challenging scenarios. After multiple outer loop iterations, the ground-truth archive accumulates diverse and challenging scenarios, testing the HRI system's strengths and limitations.

*The key idea behind the proposed algorithm is that exploiting the surrogate model with QD algorithms produces diverse – with respect to the surrogate model predictions – scenarios. Labeling these scenarios by evaluating them in a simulator and using them to retrain the surrogate model in turn improves its predictions in subsequent iterations.* Thus, the surrogate model self-improves over time by training on the diverse data generated according to its predictions.

The proposed improvements result in two versions of our algorithm: (1) Surrogate Assisted Scenario Generation (SAS), which employs a derivative-free QD algorithm, CMA-MAE, in the inner loop. (2) Differentiable Surrogate Assisted Scenario Generation (DSAS, Fig. 2), which employs a DQD algorithm, CMA-MAEGA, in the inner loop. Additional algorithm details and the pseudocode are given in App. A. Our source code is available at `https://github.com/icaros-usc/dsas`.

## 5   Domains

We consider two HRI domains from prior work: shared control teleoperation [52] and shared workspace collaboration [6] with a 6-DoF Gen2 Kinova JACO arm.

**Shared Control Teleoperation.** In shared control teleoperation, a user provides low-dimensional joystick inputs towards a goal. To aid the user, the robot attempts to infer the human goal and move towards it autonomously. An optimal policy would correctly infer the human goal and reach the goal along the shortest path. The robot solves a POMDP with the user's goal as a latent variable and updates its belief based on the human input, assuming a noisily-optimal user [52]. With hindsight optimization and first-order approximation, this results in the robot's actions being a weighted average of the optimal path towards each goal, where the weights are proportional to the respective goal probabilities. In App. H.1, we test a second robot policy that blends the user's and the robot's actions based on the robot's confidence in the user's goal [53]. Scenario parameters include both the environment, i.e., the coordinates of two goal objects, and the human actions, i.e., a set of human trajectory waypoints. To search for failures, we set the objective to be the time taken to reach the correct goal. We diversify scenarios with respect to the noise in human inputs (variation from the optimal path) and the scene clutter (distance between goals).

**Shared Workspace Collaboration.** In shared workspace collaboration, the human and the robot simultaneously execute a sequential task in a shared workspace with disjoint regions [6, 54]. An optimal robot policy would correctly infer the human's current goal and move to a different goal along the shortest path while avoiding collisions with the human. As in the teleoperation task, the robot models the human goal as a latent variable in a POMDP (but with human hand as its observation) and uses hindsight optimization and first-order value function approximations to act in real time. The robot attempts to avoid the goal intended by the human by selecting the nearest goal from a feasible set of goals different than the user's, i.e., it maps a human candidate goal to a different goal-to-go. The robot's action is a weighted average of the optimal path towards each goal-to-go, with weights proportional to the probability of the corresponding human goal.

We parameterize the scenario with three goal coordinates and set the objective to be the task completion time. We select measures based on factors that we expect to affect the team performance: object arrangement, the accuracy of inference of user's goal, and the distance required to reach the goals. We choose two sets of measures: (1) The minimum distance between goal objects and the maximum probability assigned by the robot to the wrong goal. (2) The robot path length and the total time for which the human and robot have to wait when reaching the same goal.

In this domain, we model the human as solving a softmax MDP with the maximum entropy formulation [55]. In App. H.2, we include an additional setting in the collaboration domain, where we search over both environments and human model parameters related to speed and rationality.

Additional details about the domains, the robot policy, the human policy, and the implementation are provided in App. D, App. F.1, App. F.2, and App. G respectively. In App. E, we provide example QD formulations for two additional real-world domains, but we do not run experiments on them in this paper.

## 6 Experiments

**Independent Variables.** Our two independent variables are the domain and the algorithm.

Our three domains are: (a) shared control teleoperation with distance between the goals and human variation as measures; (b) shared workspace collaboration with minimum distance between the goals and maximum wrong goal probability as measures (collaboration I); (c) shared workspace collaboration with robot path length and total wait time as measures (collaboration II).

In each domain, we compare five different algorithms: (a) **Random Search**, where we uniformly sample scenarios from the valid regions. (b) **MAP-Elites** [41], as adapted for scenario generation in previous work [2], with the additional objective regularization described in Sec. 4. (c) **CMA-MAE** [43] with objective regularization. (d) **SAS**: The proposed derivative-free version of our surrogate assisted scenario generation algorithm. We apply CMA-MAE as the derivative-free QD algorithm in the inner loop. (e) **DSAS**: The proposed differentiable surrogate assisted scenario generation with the DQD algorithm CMA-MAEGA in the inner loop.

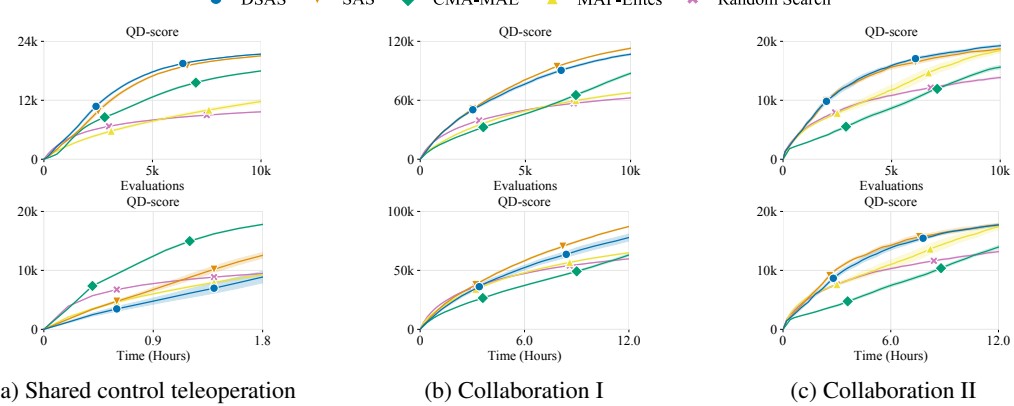

(a) Shared control teleoperation     (b) Collaboration I     (c) Collaboration II

Figure 3: QD-score attained in the three domains as a function of the number of evaluations (top) and the wall-clock time (bottom). Algorithms with surrogate models have better sample efficiency but require more wall-clock time per evaluation compared to other algorithms due to the overhead of model evaluations and model training. Plots show the mean and standard error of the mean.

**Dependent Variable:** Following previous work [2], we set QD-score [7] as the dependent variable that summarizes the quality and diversity of solutions. We compute the QD-score at the end of 10,000 evaluations, averaged over 10 trials of random search, MAP-Elites, CMA-MAE, and – because of GPU usage constraints (see App. G.3) – 5 trials of SAS and DSAS.

**Hypotheses:**

**H1.** *We hypothesize that the surrogate assisted QD algorithms SAS and DSAS will outperform CMA-MAE, MAP-Elites, and random search.* We base this hypothesis on previous work, which has shown the benefit of integrating QD with surrogate models in single-agent domains [49, 23].

**H2.** *We hypothesize that DSAS will outperform SAS.* We base this hypothesis on previous work, which has shown that DQD algorithms perform significantly better than their derivative-free counterparts [5] when the objective and measure gradients are available.

**Analysis.** A two-way ANOVA test showed a significant interaction effect ($F(8.0, 105.0) = 305.79, p < 0.001$). Simple main effects analysis on each domain showed a significant effect of the algorithm on the QD-score ($p < 0.001$). Pairwise t-tests with Bonferroni corrections showed that SAS and DSAS performed significantly better than CMA-MAE, MAP-Elites, and random search in the shared control teleoperation and collaboration I domains ($p < 0.001$). In the collaboration II domain, they outperformed CMA-MAE ($p < 0.001$) and random search ($p < 0.001$), while there was no significant difference with MAP-Elites. We attribute this to the fact that MAP-Elites can easily obtain diverse robot's path lengths by making small isotropic perturbations in the object positions (see App. H). Fig. 3 shows the QD-score as a function of the number of evaluations. We see that both SAS and DSAS achieve a high QD-score early in the search, indicating high sample efficiency.

The comparison between SAS and DSAS showed mixed results, with SAS better in the collaboration I domain ($p < 0.001$), DSAS better in the shared control teleoperation domain ($p < 0.001$), and no significance in the collaboration II domain ($p = 0.07$). Previous work [5, 43] has shown DQD algorithms improving efficiency in very high-dimensional search spaces by reducing the search from a high-dimensional solution space to a low-dimensional objective-measure space. We conjecture that this explains the significant improvement in the shared control teleoperation domain (9-dimensional solution space as opposed to 6-dimensional one in shared workspace collaboration) and we will investigate higher-dimensional domains in future work.

Furthermore, in the shared workspace collaboration domains, where scenario evaluations last a couple of minutes because of the larger task complexity, surrogate assistance showed wall-clock time efficiency (Fig. 3), unlike prior work [23] that only showed sample efficiency improvements.

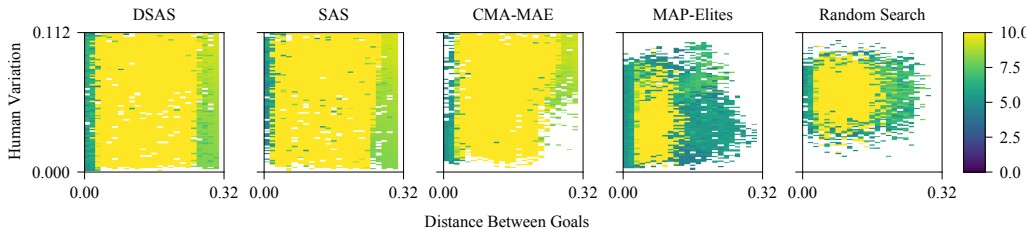

Figure 4: Comparison of the final archive heatmaps in the shared control teleoperation domain.

Fig. 4 shows heatmaps of the final archives in the shared control teleoperation domain. The heatmaps for MAP-Elites and random search match the results from prior work [2]. These heatmaps show the advantage of QD-score as a comparison metric. A higher QD-score implies the archive is filled more and with higher quality solutions. For example, SAS and DSAS find failure scenarios in the lower right corner of the archive (nearly optimal human and a large distance between the goal objects), whereas MAP-Elites fails to find failures in that region. Thus, a designer would not have information about the HRI algorithm's performance in some scenarios if the scenario generation algorithm has a low QD-score. App. H includes heatmaps in other domains and tabulates the QD-scores.

We provide additional experiments with two different settings in App. H.1 and App. H.2. Additionally, we show the effect of objective regularization in App. H.3 and examples of generated scenarios with high (as opposed to low) team performance in App. J.

**Real World Demo.** We wish to demonstrate that the generated failure scenarios are reproducible in the real world and are not just simulation artifacts. Since the algorithms that we test have been shown to work robustly in practice [6, 52, 53, 54], and the proposed approach discovers edge-case failures that are hard to find and rare in practice, a user study where users can freely interact with the system would need to involve a very large number of subjects to observe these failures. Furthermore, we would need to account for any safety concerns that arise from the unexpected robot behavior in the corner cases. We view solving these challenges as beyond the scope of this paper, and here, we only show that these failures will actually occur in the real world if users act in a certain way.

We recreate four example scenarios from the generated archives with a 6-DoF Gen2 Kinova JACO arm and by having a user reproduce the motions of the simulated human. We track the human hand position with a Kinect v1 sensor and the OpenNI package [56]. We discuss two scenarios below and the other two in App. I. We include videos of all scenarios in the supplementary material.

*Incorrect robot motion because of delayed human goal inference (Fig. 5a):* We select this scenario from the archive generated by DSAS in the collaboration II domain. In this scenario, after the human finishes working on goal G1 and the robot on G3, the robot is closer to G2 than its other remaining goal, G1. Based on the feasible goal set formulation (Sec. 5), G2 becomes the goal-to-go for human candidate goals G1 and G3. Given that the combined probability of the human going to either G1 or G3 is higher than the probability of the human going to G2, the robot moves towards G2. However, once the robot realizes that the human is actually moving to G2 as well, the robot has to move all the way back to goal G1, resulting in a significant delay.

*Long robot motion with correct human goal inference (Fig. 5c):* We additionally wish to find scenarios that result in poor team performance that is not due to incorrect inference. We select a scenario from a SAS archive in the collaboration I domain that had a low maximum wrong goal probability of 0.3. The poor performance here is caused by the interaction between the robot's policy and the object placement. As the robot moves between the two workspaces following a straight line path, it reaches a configuration close to self-collision or to joint limits. This prompts the system to re-plan and move the robot to a different configuration before continuing to move towards the goal. While re-planning ensures task completion, it induces a significant delay compared to scenarios where the goals can be reached without re-planning.

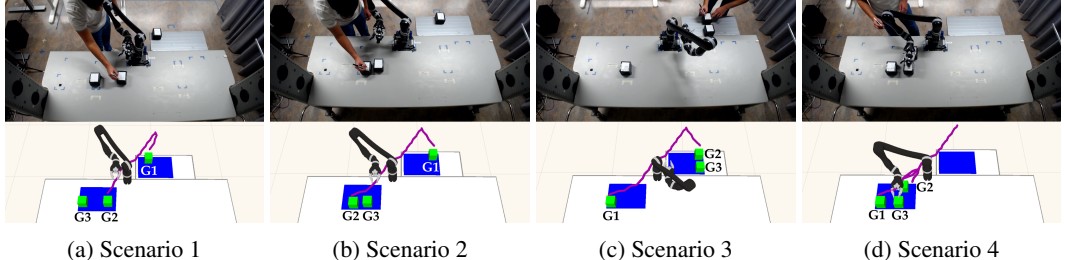

|              |              |              |              |
| :----------: | :----------: | :----------: | :----------: |
| (a) Scenario 1 | (b) Scenario 2 | (c) Scenario 3 | (d) Scenario 4 |

Figure 5: Example scenarios recreated with a real robot. The purple line shows the simulated human path. Videos of the simulated and recreated scenarios are included in the supplemental material.

# 7  Discussion

**Limitations.** Our approach scales surrogate assisted scenario generation from single-agent grid-world domains to complex human-robot interaction domains with continuous actions, environment dynamics, and object locations. However, our evaluation domains consist of objects of the same type and simple human models. We note that SAS and DSAS are general algorithms and we are excited about integrating them with more complex models of environments [51] and human actions [57, 58], as well as leveraging high-fidelity human model simulators [59, 60] to improve realism. Additionally, in our domains we were able to specify good test coverage with low-dimensional measure spaces. However, domains where we wish to obtain good test coverage over a large number of attributes would require high-dimensional measure spaces. Integrating centroidal Voronoi tessellation (CVT) based archives [61] or measure space dimensionality reduction methods [62] will allow us to apply SAS and DSAS to such domains. Furthermore, our system does not explain the reason behind the observed robot behavior in the generated scenarios, and future work will explore integrating scenario generation with methods for failure explanation [63, 64]. Finally, while we focus on a single human interacting with a single robot, we believe that our workspace occupancy-based approach for surrogate model predictions can be extended to multi-human-robot team settings.

**Implications.** We presented the SAS and DSAS scenario generation algorithms that accelerate QD scenario generation via surrogate models. Results in a shared control teleoperation domain of previous work [2] and in a shared workspace collaboration domain show significant improvements in search efficiency when generating diverse datasets of challenging scenarios.

For the first time in surrogate assisted scenario generation methods, we see improvements not only in sample efficiency but also in wall-clock time in the shared workspace collaboration domain, where evaluations last a couple of minutes. On the other hand, the additional computation in the inner loop of the surrogate assisted algorithms resulted in more time required to match and exceed the performance of the baselines in the shared control teleoperation domain, where scenario evaluations last only a few seconds. Thus, for running-time performance, we recommend surrogate assisted methods in domains with expensive evaluations, in which the additional computation in the inner loop is offset by the improvement in sample efficiency.

We additionally highlight an unexpected benefit of our system during development. When we tested the shared workspace collaboration domain, SAS and DSAS discovered failure scenarios that exploited bugs in our implementation which were subsequently fixed. For instance, some goal locations were reachable by the robot arm in the real world but unreachable in simulation because of small errors in the robot's URDF file, which prompted us to correct it.

Overall, we envision the proposed algorithms as a valuable tool to accelerate the development and testing of HRI systems before user studies and deployment. We consider this an important step towards circumventing costly failures and reducing the risk of human injuries, which is a critical milestone for widespread acceptance and use of HRI systems.

**Acknowledgments**

This work was supported by the NSF CAREER (#2145077), NSF GRFP (#DGE-1842487), and the Agilent Early Career Professor Award. One of the GPUs used in the experiments was awarded by the NVIDIA Academic Hardware Grant.

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

## A Algorithm: Differentiable Surrogate Assisted Scenario Generation

The improvements proposed in Sec. 4 result in two versions of our algorithm. In Algorithm 1, we present DSAS with the state-of-the-art DQD algorithm CMA-MAEGA in the inner loop. SAS follows a similar structure but with CMA-MAE in the inner loop. Fig. 6 shows a version of Fig. 2 with the inner loop and the outer loop split into two figures.

On each iteration of the outer loop, we initialize a new surrogate archive to store solutions that the surrogate model predicts are high performing and diverse (line 3). Then, we begin the inner loop (line 5). On line 6, we evaluate the current solution point $\theta$ with the surrogate model to obtain the predicted objective $\hat{f}$, measures $\hat{m}$, and the branching gradients $\nabla_{\hat{f}}$ and $\nabla_{\hat{m}}$. We then add the solution $\theta$ to the surrogate archive (line 8) based on the predicted evaluations, after applying the regularization penalty (line 7). Next, we generate a batch of solutions based on the branching gradients (line 9). For each solution, we sample gradient coefficients, which, combined with the gradients, produce a new candidate solution (lines 10-12). We evaluate each new candidate solution $\theta'_i$ with the surrogate model (line 13), apply the regularization penalty (line 14), and add the solution to the surrogate archive (line 15). After processing a batch, we update the search parameters of CMA-MAEGA to move the search towards maximizing the QD objective (line 17).

After completing an inner loop, we select a subset of solutions from the surrogate archive to label (line 19). For each set of scenario parameters $\theta$, we generate-and-repair a scenario (line 21), evaluate the robotic system on the scenario (line 22), update our dataset by adding the scenario labeled with the true objective $f$, measures $m$, robot occupancy grid $y_r$, and human occupancy grid $y_h$ (line 23), and finally add the scenario to our ground-truth archive (line 24). After updating the training data with newly labeled scenarios, we train the occupancy predictor for both the robot (line 27) and human (line 28), then train the surrogate model to predict the objectives and measures (line 29). The inner loop in future iterations exploits the more accurate surrogate model to produce better scenarios.

## B Surrogate Model Details

Our surrogate model follows a two-stage prediction process by first predicting the robot and the human occupancy grids given the scenario parameters as input, followed by a downstream prediction of the objective and measures.

The occupancy predictor (blue arrows in Fig. 7) consists of deconvolution layers followed by batch normalization and ReLU that treat the scenario parameters as a $1 \times 1$ image with the number of channels equal to the solution size and expand it into a $32 \times 32$ image. In the shared control teleoperation domain, there is only one occupancy grid since only the robot arm is moving. In the shared workspace collaboration domain, there are two occupancy grids (stacked into two channels) corresponding to the robot and the human motion. We pass each channel in the final output through a softmax operator and minimize the KL divergence loss between the predicted and the true occupancy grids.

The downstream predictor (red arrows in Fig. 7) consists of a fully connected network with linear layers followed by batch normalization and ReLU to extract features from the scenario parameters. It also consists of convolutional layers followed by batch normalization and leaky ReLU to extract features from the occupancy grids. We pass the features through a linear layer and minimize the mean squared error (MSE) between the predicted and the true objective and measures.

The losses for the occupancy predictor and the downstream predictor have different scales and hence, are hard to balance. Thus, we separately train both networks on data obtained from ground-truth evaluations for 100 epochs in each outer iteration using Adam [65] optimizer with a learning rate of 0.0001 and batch size of 64. We first train the occupancy predictor, freeze the weights, and then train the downstream predictor by leveraging occupancy predictions from the occupancy predictor. We implement and train the networks with the PyTorch library [66].

**Algorithm 1:** Differentiable Surrogate Assisted Scenario Generation (DSAS).

**Input:** $N$: Maximum number of evaluations, $N_{exploit}$: Number of iterations in the model exploitation phase, $\boldsymbol{\theta}_0$: Initial solution for CMA-MAEGA, $B$: Batch size for CMA-MAEGA

**Output:** Final version of the ground-truth archive $\mathcal{A}_{gt}$

1 Initialize the ground-truth archive $\mathcal{A}_{gt}$, the dataset $\mathcal{D}$, robot occupancy predictor $sm_r$, human occupancy predictor $sm_h$, objective and measure predictor $sm$

2 **while** $evals < N$ **do**

3      Initialize CMA-MAEGA with the surrogate archive $\mathcal{A}_{surr}$ and initialize solution $\boldsymbol{\theta}$ to $\boldsymbol{\theta}_0$

4      Initialize CMA-ES parameters $\boldsymbol{\mu}, \boldsymbol{\Sigma}$

5      **for** $itr \in \{1, 2, \ldots, N_{exploit}\}$ **do**

6          $\hat{f}, \boldsymbol{\nabla}_{\hat{f}}, \hat{\boldsymbol{m}}, \boldsymbol{\nabla}_{\hat{\boldsymbol{m}}} \leftarrow sm(\boldsymbol{\theta}, sm_r(\boldsymbol{\theta}), sm_h(\boldsymbol{\theta}))$

7          $\hat{f} \leftarrow \hat{f} - reg(\boldsymbol{\theta})$

8          $\mathcal{A}_{surr} \leftarrow add\_solution(\mathcal{A}_{surr}, (\boldsymbol{\theta}, \hat{f}, \hat{\boldsymbol{m}}))$

9          **for** $i \in \{1, 2, \ldots, B\}$ **do**

10              $\boldsymbol{c} \sim \mathcal{N}(\boldsymbol{\mu}, \boldsymbol{\Sigma})$

11              $\boldsymbol{\nabla}_i \leftarrow c_0 \boldsymbol{\nabla}_{\hat{f}} + \Sigma_{j=1}^{k} \left( c_j \boldsymbol{\nabla}_{\hat{\boldsymbol{m}}_j} \right)$

12              $\boldsymbol{\theta}'_i \leftarrow \boldsymbol{\theta} + \boldsymbol{\nabla}_i$

13              $\hat{f}', *, \hat{\boldsymbol{m}}', * \leftarrow sm(\boldsymbol{\theta}'_i, sm_r(\boldsymbol{\theta}'_i), sm_h(\boldsymbol{\theta}'_i))$

14              $\hat{f}' \leftarrow \hat{f}' - reg(\boldsymbol{\theta}'_i)$

15              $\mathcal{A}_{surr} \leftarrow add\_solution(\mathcal{A}_{surr}, (\boldsymbol{\theta}'_i, \hat{f}', \hat{\boldsymbol{m}}'))$

16          **end**

17          Update $\boldsymbol{\theta}, \boldsymbol{\mu}, \boldsymbol{\Sigma}$ via CMA-MAEGA update rules

18      **end**

19      $\Theta \leftarrow select\_solutions(\mathcal{A}_{surr})$

20      **for** $\boldsymbol{\theta} \in \Theta$ **do**

21          $scenario \leftarrow G(\boldsymbol{\theta})$

22          $f, \boldsymbol{m}, \boldsymbol{y}_r, \boldsymbol{y}_h \leftarrow evaluate(scenario)$

23          $\mathcal{D} \leftarrow \mathcal{D} \cup (\boldsymbol{\theta}, f, \boldsymbol{m}, \boldsymbol{y}_r, \boldsymbol{y}_h)$

24          $\mathcal{A}_{gt} \leftarrow add\_solution(\mathcal{A}_{gt}, (\boldsymbol{\theta}, f, \boldsymbol{m}))$

25          $evals \leftarrow evals + 1$

26      **end**

27      $sm_r.train(\mathcal{D})$

28      $sm_h.train(\mathcal{D})$

29      $sm.train(\mathcal{D}, sm_r, sm_h)$

30 **end**

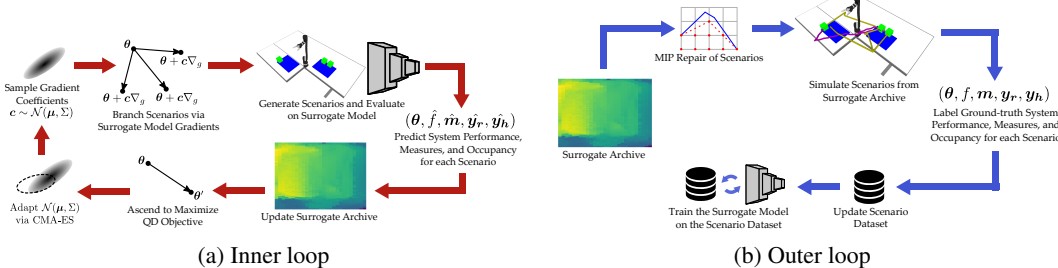

(a) Inner loop                       (b) Outer loop

Figure 6: An overview of our proposed differentiable surrogate assisted scenario generation (DSAS) algorithm for HRI tasks. (a) The inner loop, where a QD algorithm exploits a surrogate model to obtain scenarios that are predicted to be challenging and diverse. (b) The outer loop, where the scenarios are labeled after repairing and evaluating them in a simulator. This data is leveraged to train and improve the surrogate model for subsequent iterations.

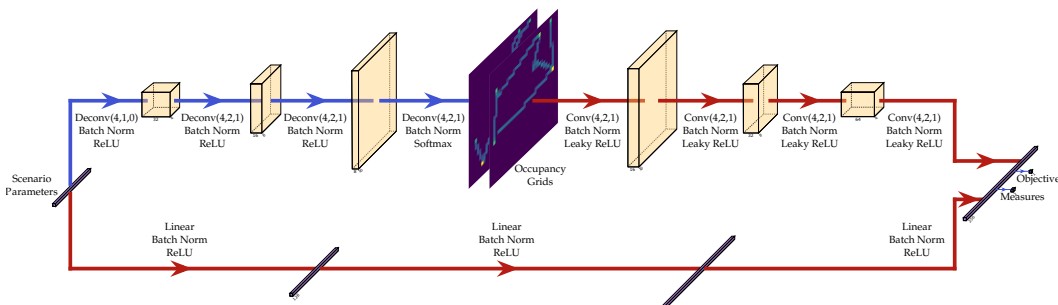

Figure 7: Architecture of the surrogate model including the occupancy predictor (**blue arrows**) and the downstream predictor (**red arrows**).

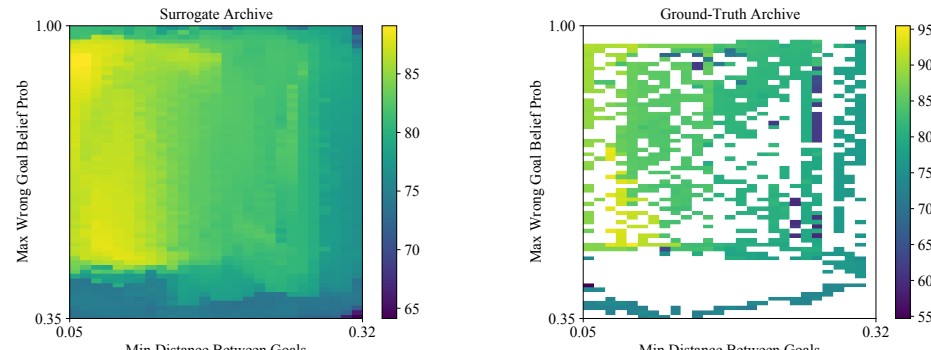

Figure 8: Comparison between the surrogate archive (left) after an inner loop and the corresponding ground-truth archive (right) after evaluating the solutions in the surrogate archive.

## B.1 Evaluating the Surrogate Model Predictions

We evaluate the predictions of the surrogate model similar to DSAGE [23] by taking the dataset generated in one trial of an algorithm and treating it as the test set for the trained surrogate model from another trial of the same algorithm. Table 1 shows the mean absolute error (MAE) in all three domains for the surrogate models trained as a part of both DSAS and SAS. Note that *Measure 1* and *Measure 2* columns in the table correspond to the respective measures in each domain described in Sec. 5.

The surrogate model is able to accurately predict the measures in the shared control teleoperation domain since they can be calculated directly from the solution and do not depend on the robot policy. In contrast, measures that depend on the robot policy such as the maximum wrong goal probability (*Measure 2* in the collaboration I domain) have a comparatively higher error, with predictions being off by around 9% on average.

Table 1: Mean absolute error of the objective and measure predictions by the surrogate models.

| Domain | DSAS | | | SAS | | |
|---|---|---|---|---|---|---|
| | Objective MAE | Measure 1 MAE | Measure 2 MAE | Objective MAE | Measure 1 MAE | Measure 2 MAE |
| Shared Control Teleoperation | 0.35 | 0.01 | 0.01 | 0.64 | 0.02 | 0.01 |
| Collaboration I | 3.41 | 0.02 | 0.09 | 3.47 | 0.02 | 0.08 |
| Collaboration II | 3.22 | 0.27 | 0.56 | 3.39 | 0.29 | 0.59 |

Furthermore, we observe that the percentage of predictions landing in their true archive cell is only around 2-4% in all domains. Nonetheless, the predictions are close to their true archive cell as evident in the MAEs. We also confirm this by computing the average Manhattan distance between the predicted archive cell and the true archive cell for each solution. In the shared control teleoperation domain, the average Manhattan distance was 6.48 and 11.26 for DSAS and SAS respectively. The average Manhattan distances for DSAS and SAS were 11.26 and 9.88 in the collaboration I domain, and 6.89 and 7.20 in the collaboration II domain, indicating that the predicted archive cells are only a few cells away from the true archive cells on average.

Thus, despite inaccuracies in placing the solutions into their true archive cells, the solutions in the surrogate archive are diverse with respect to the true measure functions. Hence, when these solutions are evaluated, they occupy different parts of the ground-truth archive and rapidly improve the QD-score. Fig. 8 shows the surrogate archive after one inner loop and the corresponding ground-truth archive obtained by evaluating the solutions in the surrogate archive in the collaboration I domain.

## C   Mixed Integer Program for Repairing Scenarios

To ensure that the objects in the scenario generated by QD search satisfy the object arrangement constraints in the shared workspace collaboration domain, we adopt a generate-then-repair strategy. We formulate a mixed integer program (MIP) with constraints to ensure that the objects in the scenario are inside the workspace boundaries and not in collision with each other. Since we wish the repaired scenario to be as close as possible to the generated scenario, we set the MIP objective to be the $L2$ distance between the original position and the repaired position of the objects. The quadratic objective makes the MIP a mixed integer quadratic program (MIQP).

### C.1   Variables and MIP Objective

We treat the $x$ and $y$ coordinates of each object as the MIP variables. Let $x'_i$ and $y'_i$ be the coordinates of object $i$ in the generated scenario and let $x_i$, and $y_i$ be the corresponding coordinates after MIP repair. We set the objective to be:

$$\min \Sigma_i (x_i - x'_i)^2 + (y_i - y'_i)^2 \tag{1}$$

### C.2   Constraints

Let $x_r^{(min)}$, $x_r^{(max)}$, $y_r^{(min)}$, and $y_r^{(max)}$ be the minimum and maximum allowed $x$ and $y$ values respectively for objects in a rectangular workspace region $r$. For each workspace region, we need to construct a binary variable $z_{ir}^{(in)}$, which resolves to true if object $i$ occupies workspace $r$. We create four auxiliary decision variables $z_{ir}^{(up)}$, $z_{ir}^{(dn)}$, $z_{ir}^{(lt)}$, and $z_{ir}^{(rt)}$, representing the four boundary constraints of the rectangle. Specifically, $z_{ir}^{(up)}$ represents if object $i$ occupies $\langle x_i, y_i \rangle$ coordinates below the top of the bounding rectangle for region $r$. The variables $z_{ir}^{(dn)}$, $z_{ir}^{(lt)}$, and $z_{ir}^{(rt)}$ satisfy the same conditions for the bottom, left, and right of the bounding rectangle, respectively. For each pair of object $i$ and region $r$, we add the following constraints to the MIP to resolve the decision variables:

$$x_r^{(min)} \leq x_i + \infty(1 - z_{ir}^{(lt)}) \tag{2}$$

$$x_i \leq x_r^{(max)} + \infty(1 - z_{ir}^{(rt)}) \tag{3}$$

$$y_r^{(min)} \leq y_i + \infty(1 - z_{ir}^{(dn)}) \tag{4}$$

$$y_i \leq y_r^{(max)} + \infty(1 - z_{ir}^{(up)}) \tag{5}$$

In the above constraints, the $\infty$ value represents a sufficiently large constant (e.g., the maximum of the width and height of a global bounding box) that causes the constraint to always be satisfied. For example, in Eq. 2, the inequality is always satisfied if the binary decision variable $z_{ir}^{(lt)}$ is false as

we do not need to put any constraints if we do not occupy region $r$ with object $i$. However, if the variable is true, we require that the coordinate $x_i$ is to the right of the $x$-boundary $x_r^{(min)}$. We create an equivalent constraint for the remaining three rectangular constraints (see Eq. 3-Eq. 5).

Finally, we add a constraint that resolves the decision variable $z_{ir}^{(in)}$ to true if all four rectangular constraints hold:
$$4 \leq z_{ir}^{(lt)} + z_{ir}^{(rt)} + z_{ir}^{(dn)} + z_{ir}^{(up)} + \infty(1 - z_{ir}^{(in)}) \tag{6}$$

Once again, if $z_{ir}^{(in)}$ is false, the inequality holds as we do not need to satisfy the rectangle inclusion constraints if our object $i$ is not in region $r$. Otherwise, all four inclusion variables must be true, by summing to four, to indicate that the object $i$ occupies region $r$.

We then add an additional constraint to ensure that each object occupies at least one region:
$$\forall i, \Sigma_r z_{ir}^{(in)} >= 1 \tag{7}$$

Next, we ensure that all pairs of objects in the scene do not overlap. To do this, we constrain the bounding boxes of each object to not overlap. Let $a_i$ be half of the side length of the bounding box of object $i$. There are four ways a pair of objects with axis-aligned bounding rectangles can avoid overlapping: object $i$ is left of object $j$, object $i$ is right of object $j$, object $i$ is above object $j$, or object $i$ is below object $j$. We create indicator variables representing these conditions as $c_{ij}^{(lt)}$, $c_{ij}^{(rt)}$, $c_{ij}^{(up)}$, $c_{ij}^{(dn)}$, respectively. Next, we add the following constraints to the MIP to correctly set the collision indicator variables:
$$(x_i + a_i) \leq (x_j - a_j) + \infty(1 - c_{ij}^{(lt)}) \tag{8}$$
$$(x_j + a_j) \leq (x_i - a_i) + \infty(1 - c_{ij}^{(rt)}) \tag{9}$$
$$(y_i + a_i) \leq (y_j - a_j) + \infty(1 - c_{ij}^{(dn)}) \tag{10}$$
$$(y_j + a_j) \leq (y_i - a_i) + \infty(1 - c_{ij}^{(up)}) \tag{11}$$

If there is no collision between $i$ and $j$, at least one of the four indicator variables must be true. Hence, we set an additional constraint to ensure no collision:
$$\forall_{i,j}, c_{ij}^{(lt)} + c_{ij}^{(rt)} + c_{ij}^{(dn)} + c_{ij}^{(up)} >= 1 \tag{12}$$

We solve the MIP problem with IBM's CPLEX optimization library [67].

## D Domains

The following subsections provide a brief description of the search space, objective, and measure functions in our domains.

### D.1 Shared Control Teleoperation

A teleoperation task involves a user providing joystick inputs to a robot arm with the intention of reaching a goal in the environment (Fig. 9a). It is generally hard for users to teleoperate a 6-DoF robot arm to the correct configuration [52]. Thus, in shared control teleoperation, the robot attempts to infer the human goal from a set of candidate goals by observing the low-dimensional joystick inputs provided by the user.

Following the shared control teleoperation framework from previous work [52], the robot solves a POMDP with the user's goal as a latent variable while it updates its belief about the goal based on the human input trajectory assuming a noisily-optimal user. To enable real-time decision-making, the robot performs hindsight optimization to approximate the POMDP and assumes a first-order approximation of the value function. This results in the robot's actions being a weighted average of the optimal path towards each goal, where the weights are proportional to the respective goal probabilities.

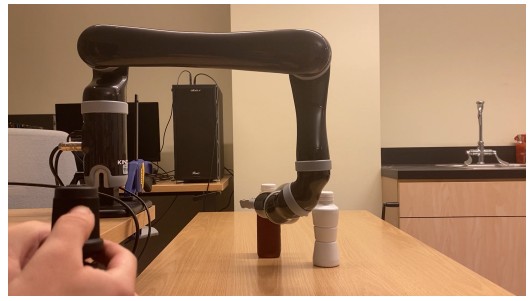
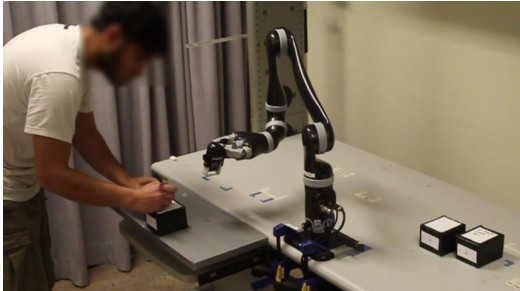

| (a) Shared Control Teleoperation | (b) Shared Workspace Collaboration |

Figure 9: Example scenarios from the two domains being executed in the real world (figure for shared control teleoperation taken from prior work [2]).

To formalize the scenario generation problem in the shared teleoperation domain, we follow the QD formulation of prior work [2, 3]. The environment parameters are the positions of the two goal objects in a bounded workspace, constrained to be reachable by the robot arm. The simulated human provides a trajectory of joystick inputs towards their goal object, parameterized by a set of waypoints. The human model parameters are disturbances to these waypoints. The scenario parameters $\theta$ include the environment and human model parameters. The objective function $f$ in the QD search is the time taken to reach the correct goal, with a maximum time limit of 10 seconds if the robot fails to reach the goal. The search aims to find scenarios that are diverse with respect to the noise in human inputs and the scene clutter, thus the measures $m$ are the human variation from the optimal path and the distance between goals.

### D.2 Shared Workspace Collaboration

We consider a package labeling task (Fig. 9b), which instantiates the human-robot shared workspace collaboration domain of previous work [6, 54]. The human and the robot have different actions, i.e., the human labels a package while the robot presses a stamp, and they share a set of goals, i.e., boxes to perform the task. The human and the robot cannot work simultaneously on the same object and the task finishes when all boxes are labeled and stamped.

We assume that the human picks a label for an object from a starting point and moves towards that object. Different boxes require different labels, thus we model the human as attempting to reach the box corresponding to the label they picked up, regardless of the robot's actions. On the other hand, the robot can switch its goal while moving, since stamping can be performed on any goal object with the same tool. This domain is more complex than the shared control teleoperation task because it includes manipulating a sequence of objects, rather than reaching a single object, and the objects are in disjoint workspace regions.

As in the shared control teleoperation task, the robot reasons over the human goal by treating the human as noisily-optimal. However, unlike in shared control teleoperation, the robot attempts to avoid the goal intended by the human.

The scenario parameters consist of the locations of three goal objects in a larger, disconnected workspace. We set the workspace boundaries to the quadrants of the L-shaped table in Fig. 1 that are reachable by both the human and the robot arm. We model the human as moving to their goal while avoiding obstacles by solving a softmax MDP. The objective $f$ is again the time to task completion since we wish to find challenging scenarios.

We choose two sets of measures $m$ described below:

**Minimum distance between goal objects and maximum wrong goal probability:** We adopt the minimum distance measure from the shared control teleoperation domain in previous work [2]. Furthermore, one of the failure scenarios found in that work was caused by incorrect inference of

the human goal by the robot. Thus, we set as our second measure the maximum probability that is assigned to the wrong goal by the robot during the task, to search for potential failures in which the robot actually infers the human goal correctly.

**Robot path length and total wait time:** In the shared workspace collaboration task that we consider, there are two main sources of delay: the robot needing to move across the two workspaces to reach different goals, and the wait time caused due to both the human and the robot wanting to work on the same goal. Hence, we choose the path length of the robot and the total wait time as the two measures to see how the team performance changes as these are varied.

## E  Additional Example QD Formulations

While we consider two domains in our paper: shared control teleoperation and shared workspace collaboration, the QD formulation and our algorithms can be extended to other HRI domains as well. We discuss example QD formulations for two alternative domains below.

**Robot navigation around pedestrians [68, 69].** The robot's objective here would be to minimize the time required to reach its goal while avoiding collisions with pedestrians. To find failures, the QD objective could be to maximize the time taken by the robot to reach the goal. Measures could include the number and size of obstacles in the scene, the robot's goal location, the minimum distance between the mobile robot and the pedestrians during navigation, the average speed of pedestrians, the curvature of the pedestrian trajectories, the total number of seconds the robot remained idle during the task, etc.

**Assistive Feeding [70].** The robot's objective here would be to successfully transfer all bites on a plate to the user. To find failures, the QD objective could be to maximize the number of failed bite transfers. Measures could include aspects of the user's capability (e.g., how far the user can reach to receive a bite, how long the user takes to receive a bite), physical user characteristics such as their height, time spent by the user being idle waiting for a bite, etc.

## F  Human and Robot Policies

### F.1  Robot Policy

We adopt the robot policy defined in prior HRI works [52, 6] that introduced the domains considered in this paper. In both domains in this paper, the robot solves a POMDP with human goal as the latent variable. As in prior work [52], the robot assumes that the human is stochastically optimal and updates its belief based on observed human actions. It performs hindsight optimization to calculate the values and update the belief in real-time, followed by a first-order approximation to select the optimal action that maximizes the Q-value. In both domains, we follow the cost function definition in the corresponding prior work [52, 6], which makes the resulting optimal value function proportional to the distance to the goal and the optimal policy a straight line.

We briefly discuss the specifics of the robot policy in the two domains below. In both domains, the robot action is computed as the twist that should be applied to its end effector, which is then converted to the required joint velocities by inverse kinematics computation.

### F.1.1  Shared Control Teleoperation

In shared control teleoperation, the human provides an input action to the robot. The Q-value of this action is defined as the sum of the cost incurred while executing the action and the value at the new position after action execution. The robot's belief is then updated based on the difference between the value and the Q-value at the current position corresponding to each goal.

Hindsight optimization followed by first-order approximation results in the robot's assistive action being a weighted average of the straight-line paths to each goal, weighted by the corresponding probabilities assigned to them in the belief.

In App. H.1, we consider a different robot policy called policy blending [53]. The robot fully follows the user inputs while updating its belief like before. Once the probability assigned to a goal is higher than a threshold, the robot takes over and moves to the predicted goal.

### F.1.2 Shared Workspace Collaboration

In shared workspace collaboration, the human acts independently. Hence, we maintain two sets of value functions - one for the human and one for the robot. We calculate the human Q-value as the sum of the cost of executing the current action and the value at the new position after action execution, similar to the shared control teleoperation domain. The robot's belief is updated based on the difference between human value and human Q-value at the current position corresponding to each goal.

We track the constraints on the robot's goals with the feasible goal-set formulation from prior work [6]. For each potential human goal, the robot maintains a set of goals that it has not worked on and is different from the human goal. The goal set can be empty for some candidate human goals if the robot has finished working on all other goals. The robot then treats all the goals that it has not worked on as the feasible goal set corresponding to that human goal. For action calculation, the robot creates a mapping from each human goal to a corresponding goal-to-go, which is the goal with minimum value (the closest goal) in the corresponding feasible goal set.

The robot's action is based on the robot's value functions. Since we assume that the robot acts optimally, we do not explicitly calculate these values and simply assume a straight-line path to each goal. Hindsight optimization followed by first-order approximation once again results in the robot's action being a weighted average of optimal actions towards each goal-to-go.

Specifically, let $b(g)$ be the probability assigned to goal $g$ and let $F(g)$ be the goal-to-go corresponding to human goal $g$. Then, the weight corresponding to goal $g'$ is given by $\Sigma_{g:F(g)=g'} b(g)$.

### F.2 Human Policy

### F.2.1 Shared Control Teleoperation

In shared control teleoperation, we search for human policy parameters in the form of noise added to the waypoints from the starting location to the intended goal location. The human policy keeps track of the waypoints and computes the waypoint-to-go and the corresponding velocity based on the current position of the robot arm.

### F.2.2 Shared Workspace Collaboration

In shared workspace collaboration, the human moves independently towards the goal and avoids obstacles on the way. We model the human policy through a softmax MDP whose values are pre-computed before simulating the scenario.

First, we discretize the space in which the human can move into a grid with cell sizes equal to the size of the goal object so that each goal is in one cell. We treat these cells as the states of the MDP and allow the human to move to any neighboring cell, receiving a reward of either $-0.01$ for moving to an orthogonally adjacent cell, $-0.01\sqrt{2}$ for moving to a diagonally adjacent cell, $-1$ for moving into an obstacle, or $1$ for moving into a goal cell. We set the discount factor to $0.9999$ and perform softmax value iteration [55] with a softmax temperature of $0.001$ to compute the Q-values for each state-action pair.

Since we have three goals in a scenario, we compute three sets of Q-values, one corresponding to each goal. Each value iteration instantiation treats the scenario's other goals as obstacles.

During simulation, the human policy converts the current location of the human into the grid cell it belongs to, chooses the next grid cell based on the Q-values corresponding to the current goal, and returns the velocity required to move to the center of the next cell.

In App. H.2, we consider a new setting in which we search over two human model parameters: the inverse of softmax temperature (higher values result in a more rational human) and a multiplier to the velocity (higher multiplier makes the human move faster).

# G   Implementation Details

We implement surrogate assisted scenario generation in a server-client framework. The server simulates a given scenario in OpenRAVE [71] while the client executes QD search to generate new scenarios.

## G.1   Scenario Simulation

We adapt the scenario simulation code from the open-source implementation of shared autonomy via hindsight optimization [72] to include the feasible goal set formulation for the shared workspace collaboration domain (described in App. F.2) and to simulate generated scenarios instead of a fixed one.

We start a flask server that waits for the client to run QD search and send solutions to evaluate. Once we receive a candidate solution, we pass it through the MIP solver and instantiate the objects, the robot, and the human in the OpenRAVE simulator.

We discretize the simulation into *ticks*, with each tick being divided into three phases that are executed in sequence: human action selection, robot action selection, and environment simulation. Human action selection and robot action selection follow the policy given in App. F.2 and App. F.1 respectively. In the environment simulation phase, the actions are executed, moving the human and the robot to a new state.

The shared control teleoperation task executes these phases in a loop until the robot reaches the intended human goal or the time limit of 10 seconds is reached.

Since the shared workspace collaboration task consists of multiple steps, the human and the robot policies are wrapped into state machines. The human state machine has five states: a) *moving to a goal*; b) *waiting for space*; c) *working on a goal*; d) *resetting*; e) *done*. The human is initially in *moving to goal* state and simply selects actions according to the human policy. Once a goal is reached, the human waits till the goal is free to work on (*waiting for space*) and then starts working on the goal (*working on a goal*). Once the work is complete, the human switches to the terminal state, *done*, if that was the last goal or moves back to the initial position (*resetting*). To simulate working on the goal and moving back to the initial position, we simply pause the human for a specified amount of time. After the reset, the human starts moving to the next goal (*moving to goal*).

The robot state machine has six states: a) *moving to a goal*; b) *replanning*; c) *waiting for space*; d) *working on a goal*; e) *resetting*; f) *done*. The state transitions are similar to those of the human state machine, except for the *moving to a goal* state. Since the robot can get into configurations close to self-collision or joint limits when following a straight line path, it needs to replan back to the start before moving again. We simulate this by switching to *replanning* state, moving the robot back to its initial position, and then switching back to *moving to a goal* state.

The shared workspace collaboration task ends either after 100 seconds or after both the human and the robot reach the *done* state.

## G.2   QD Search

We implement QD search on the client by modifying the pyribs library [73] and the open-source code for DSAGE [23] to match Algorithm 1.

We implement the inner loop through a pyribs *scheduler* that interfaces a QD algorithm via two functions: *ask*, which outputs candidate solutions from the algorithm, and *tell*, which accepts the corresponding objective and measures, adds them to the archive, and updates the algorithm param-

eters. The scheduler interfaces CMA-MAEGA and CMA-MAE for DSAS and SAS respectively. The inner loop runs fully on the client, exploiting the surrogate model described in App. B.

We then select a set of solutions from the surrogate archive and send it to the simulation server for evaluation. The objective and measures obtained from the simulation are returned by the server, which we add to the ground-truth archive and the dataset.

For baselines, we use the existing implementation of CMA-MAE and MAP-Elites in pyribs. Additionally, for ease of execution, we implement Random Search similar to a QD algorithm in the pyribs framework. It simply returns a batch of uniformly randomly sampled candidate solutions whenever requested. Since these baselines do not leverage a surrogate model, the candidate solutions are always sent to the simulation server for evaluation.

To include objective regularization, we maintain two archives, the *final archive* that retains solutions maximizing the unregularized objective, and the *training archive*, which maintains scenarios that maximize the regularized objective to guide the QD search. The pyribs scheduler interfaces with the *training archive*, while solutions are directly added to the *final archive*. For surrogate assisted algorithms, the surrogate archive acts as the *training archive* while the ground-truth archive acts as the *final archive*.

We include the search details specific to the domains below.

### G.2.1   Shared Control Teleoperation

In shared control teleoperation, we search over the $\langle x, y \rangle$ coordinates of two goal objects and five noise variables that define the human path towards the goal, creating a 9-dimensional search space.

We define the measures as the distance between the goals $\sqrt{(x_1 - x_2)^2 + (y_1 - y_2)^2}$, and the variation in human input $\sqrt{\Sigma_{i=1}^{5} \boldsymbol{\theta}_{h,i}}$, where $\boldsymbol{\theta}_h$ refers to the five noise parameters in the generated solution. Following prior work [2], we assume the ranges of the measures to be $[0, 0.32]$ for the distance and $[0, 0.112]$ for variation, and create an archive with $25 \times 100$ cells.

We adopt the hyperparameters for MAP-Elites from prior work [2], setting the standard deviation of perturbation, $\sigma$, to 0.01 for parameters corresponding to the goal coordinates and 0.005 for those corresponding to the human noise. For CMA-MAE, SAS, and DSAS, we set the initial standard deviation for CMA-ES, $\sigma_0$, to 0.01, archive learning rate, $\alpha$, to 0.1, and minimum acceptance threshold, $min_f$, to 0. We set all other hyperparameters to their default values defined in pyribs.

### G.2.2   Shared Workspace Collaboration

In shared workspace collaboration, we search over the $\langle x, y \rangle$ coordinates of three goal objects, creating a 6-dimensional search space.

The four measure functions in our experiments are defined as follows:

1. Minimum distance between goal objects (archive range $[0.05, 0.32]$; discretized into 27 archive cells): $\min_{i \neq j} \sqrt{(x_i - x_j)^2 + (y_i - y_j)^2}$

2. Maximum wrong goal probability (archive range $[0.35, 1]$; discretized into 65 archive cells): Let $b^{(max)}(t)$ be a function that returns the highest probability assigned by the robot to a goal other than the true human goal at time $t$. Maximum wrong goal probability is defined as the maximum value attained by $b^{(max)}(t)$ during the scenario: $\max_t b^{(max)}(t)$.

3. Robot path length (archive range $[1, 5]$; discretized into 20 archive cells): Let the robot's trajectory in the scenario be a function $\tau : [0, 1] \to \mathbb{R}^2$, with $\tau(0)$ and $\tau(1)$ denoting the coordinates of the start and end-points respectively. The robot path length is defined as the length of this trajectory: $\int_0^1 \|d\tau\|_2$.

4. Total wait time (archive range $[0, 5]$; discretized into 50 archive cells): Let $w(t)$ be a function that returns 1 when either the robot or the human state machine is in *waiting for space*

Table 2: QD-score at the end of 10,000 evaluations.

|  | Shared Autonomy | Collaboration I | Collaboration II |
|---|---|---|---|
| DSAS | $\mathbf{21,400.33 \pm 45.91}$ | $106,874.93 \pm 844.00$ | $\mathbf{19,261.95 \pm 182.57}$ |
| SAS | $21,043.49 \pm 40.08$ | $\mathbf{112,962.22 \pm 572.96}$ | $\mathbf{18,733.82 \pm 182.40}$ |
| CMA-MAE | $17,972.31 \pm 74.71$ | $87,399.75 \pm 1,085.14$ | $15,612.29 \pm 284.34$ |
| MAP-Elites | $11,757.84 \pm 358.31$ | $67,731.48 \pm 576.30$ | $\mathbf{18,435.18 \pm 398.87}$ |
| Random Search | $9,647.24 \pm 24.94$ | $62,376.62 \pm 200.68$ | $13,856.14 \pm 156.67$ |

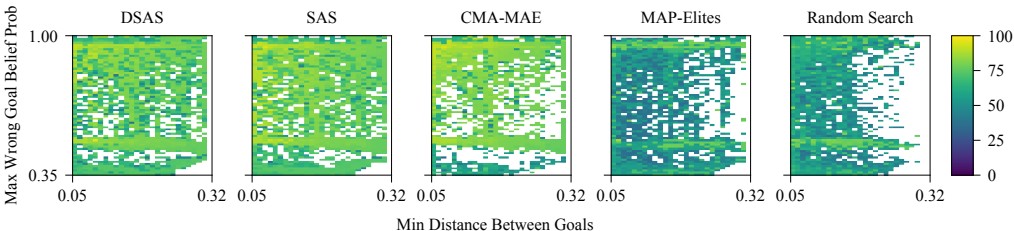

Figure 10: Comparison of the final archive heatmaps in the collaboration I domain.

state (see App. G.1) and 0 otherwise. Total wait time is defined as $\int_0^T w(t)dt$, where $T$ is the total scenario time.

Note that we approximate the integrals with discrete sums of the corresponding values at each simulation tick.

We tuned the initial standard deviation for CMA-ES, $\sigma_0$, in the case of CMA-MAE, SAS, and DSAS and set it to 1. We also tuned the perturbation standard deviation, $\sigma$, for MAP-Elites and set it to 0.1. We set $\alpha = 0.1$, $min_f = 0$, and all other hyperparameters to the default values provided in pyribs.

In the new setting described in App. H.2, we add two additional parameters to the search: the inverse of softmax temperature (higher values result in a more rational human) and the coefficient of velocity (higher coefficient makes the human move faster). We limit these parameters to ensure that the scenarios are not bottlenecked by an unrealistically slow or irrational human.

### G.3 Computational Resources

Experiments for this paper were run on two local machines and a shared high-performance cluster. The local machines had AMD Ryzen Threadripper with a 64-core (128 threads) CPU and an NVIDIA GeForce RTX 3090/RTX A6000 GPU. 16 CPU cores were allocated for each run on the cluster. One V100 GPU was additionally allocated for runs with a surrogate model.

The total time for each run was between 2 to 10 hours in the shared control teleoperation domain and between 12 - 24 hours in the shared workspace collaboration domain.

## H Additional Results

We tabulate the results from our experiments in Table 2. We also show the final archives in the collaboration I (Fig. 10) and collaboration II (Fig. 11) domains.

We observe that the archives generated by DSAS and SAS are more densely packed compared to other algorithms in collaboration I. In collaboration II, we see that CMA-MAE, SAS, and DSAS find fewer solutions in the bottom left part of the archive compared to MAP-Elites and random search, but find more and higher quality solutions in other parts of the archive which requires placing the goals in multiple regions. This is due to the bottom left part mostly corresponding to all goal objects

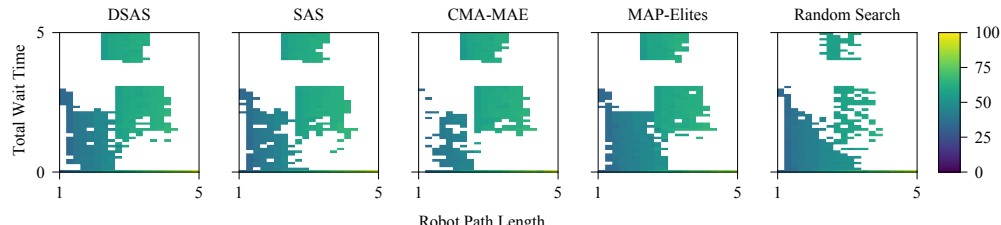

Figure 11: Comparison of the final archive heatmaps in the collaboration II domain.

Table 3: QD-score at the end of 10,000 evaluations.

|  | Teleoperation (Policy Blending) | Collaboration I (Human Policy Search) |
|---|---|---|
| DSAS | $\mathbf{41,249.88 \pm 205.56}$ | $106,573.18 \pm 1,461.34$ |
| SAS | $40,726.15 \pm 300.61$ | $\mathbf{120,789.83 \pm 1,378.82}$ |
| CMA-MAE | $33,797.07 \pm 1,455.82$ | $\mathbf{120,687.02 \pm 2,959.76}$ |
| MAP-Elites | $24,151.97 \pm 836.97$ | $81,006.04 \pm 2,483.23$ |
| Random Search | $19,850.68 \pm 184.97$ | $65,513.84 \pm 334.12$ |

in one workspace region and low wait time which allows MAP-Elites to add small perturbations to the goal locations and easily obtain different robot path lengths. However, maintaining a high wait time (the top part of the archive) while adjusting the robot path lengths is harder since even small perturbations in goal positions could make it easy for the robot to infer the human's goal and go to a different goal.

### H.1 Additional Setting: Shared Control Teleoperation with Policy Blending

The QD formulation for scenario generation is independent of the robot and human policies. Here, we show scenario generation with a new robot policy, policy blending (App. F.1), in the shared control teleoperation domain without any modifications to the QD hyperparameters or the surrogate model architecture.

Table 3 shows the QD-score at the end of 10,000 evaluations. We see that the surrogate assisted algorithms outperform other algorithms, showing that these algorithms can work across multiple robot policies. Note that the maximum time for a scenario was set to 20s, so the QD-scores are around twice as large as in the main shared teleoperation experiments.

### H.2 Additional Setting: Shared Workspace Collaboration with Human Policy Search

In the main shared workspace collaboration experiments, the scenario was only parameterized by object locations. However, as described in our problem formulation, scenario parameters can also include parameters of the human model. Here, we perform an additional experiment in which we search for human model parameters in addition to the object locations to find failures in the collaboration I domain. We add two more scenario parameters related to human speed and rationality as described in App. G.2 and run the QD algorithms with no other changes to the hyperparameters.

We tabulate the QD-scores in Table 3. We see a small increase in the QD-scores of all algorithms compared to the main experiments (Table 2), since the QD search can now control the human policy to cause failures. Surprisingly, CMA-MAE performs similar to SAS. We hypothesize that this is caused by the sensitivity of the scenario outcomes to the human model parameters: Changes to human speed or rationality affect the human trajectory much more than changes to goal locations. Hence, predicting the trajectory and scenario outcomes is much harder in this setting compared to

the main experiments. Thus, CMA-MAE, a model-free QD algorithm, performs as well as SAS and outperforms DSAS.

However, the failures broadly fell into the same categories as those found in the main experiment. We hypothesize that this results from the bounds of the human policy parameters. Rational and fast human actions allow the robot to accurately predict the human's goal, leading to fast scenario completion. On the other hand, we have set the bounds on the parameters to not allow QD search to make the human unrealistically slow or irrational. Hence, the failures found in this experiment are similar to those found with a fixed human policy.

### H.3 Ablation: Effect of Objective Regularization

In Sec. 4, we proposed objective regularization as a way to guide QD search towards valid workspace configurations. While objective regularization benefits general QD search, we note that surrogate assisted methods like DSAGE inherit additional benefits. As the surrogate model makes predictions for all possible scenarios, and not only scenarios satisfying the workspace constraints, the QD search that exploits the surrogate model can move towards high-magnitude inputs in invalid regions of the scenario space when these inputs result in high objective values. Objective regularization helps prevent QD algorithms from exploiting errors in the surrogate model at extreme regions of the scenario parameter space.

To test the effect of objective regularization on performance, we choose the collaboration I domain and run 10 trials of DSAS, SAS, CMA-MAE, and MAP-Elites without objective regularization. Hence, due to numerical errors resulting from exploiting errors in the surrogate model, none of the SAS or DSAS runs without objective regularization could be completed.

We compare the results of MAP-Elites and CMA-MAE runs with their corresponding runs from the previous section that included objective regularization. Pairwise t-tests showed that MAP-Elites performed similarly with and without regularization, while CMA-MAE performed significantly worse without objective regularization ($t = -7.08, p < 0.001$). We attribute this to the fact that perturbations of existing solutions in MAP-Elites are not guided by the objective values. On the other hand, CMA-MAE guides the search based on the objective improvements of the sampled solutions; hence objective regularization has a significant effect on performance.

## I  Additional Real World Scenarios

*Incorrect human goal inference with limited effect on robot motion (Fig. 5b):* We select a scenario from the archive generated by SAS with a relatively average scenario time of 77s and a very high maximum wrong goal probability of 0.9.

The human finishes working on G1 and the robot on G2. As the human moves towards G2, the robot incorrectly thinks that the human is moving to G3, which is near the optimal path to G2, causing the robot to slow down in anticipation of the human motion. After the human reaches G2, the robot continues moving to G3. Hence, the incorrect prediction does not affect the overall scenario time much.

*Long wait time due to both teammates needing to work on the same goal (Fig. 5d):* Finally, we select a scenario from a DSAS archive in the collaboration II domain that has a high human and robot wait time.

This scenario was simple, albeit unanticipated. The human goes to G1, followed by G2, while the robot goes to G2, followed by G1. The team coordinates smoothly until both agents need to work on G3 to finish the task, causing a delay.

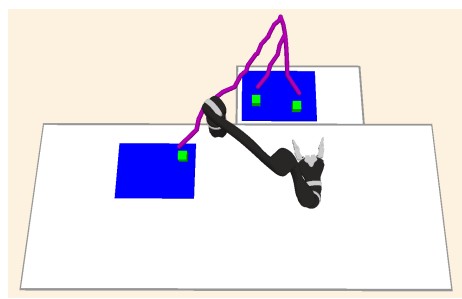 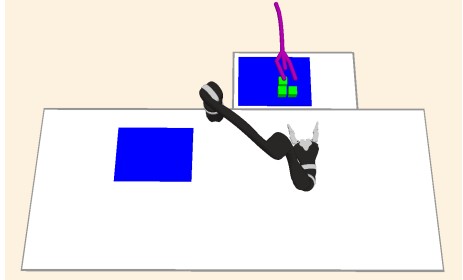

(a) High Team Performance Scenario 1    (b) High Team Performance Scenario 2

Figure 12: Examples of scenarios with high team performance. The purple line shows the simulated human path.

## J  Scenarios with High Team Performance

In addition to finding failures, QD scenario generation can also find scenarios that are ideal for human-robot collaboration. As an example, we modified the objective function in the collaboration I domain to $100 - T$, with $T$ being the scenario completion time that has a maximum value of 100s. We ran SAS, which performed the best in this domain, and visualized example scenarios. We found two main types of success scenarios:

*Objects placed far apart to avoid confusion (Fig. 12a):* The first type of success involved placing the objects far apart to allow accurate goal inference. However, placing them too far would require the human and the robot to move a lot, delaying completion. This scenario balanced these trade-offs, leading to a relatively short robot path length of 1.7m, a low maximum wrong goal probability of 0.4, and a fast completion time of 38s. The resulting goal completion order also avoided the failure found in Fig. 5d.

*Objects placed close together to quickly change goals (Fig. 12b):* The second type of success ignored making goal inference easier but instead made it easier for the robot to correct itself if required. Since the goals are close to each other, the robot can start moving towards them irrespective of human actions. Once the human starts working on a goal, the robot can quickly switch to a different goal. Despite having a high maximum wrong goal probability of 0.8, this scenario only took 31s to complete.

