# OpenReview forum: "Surrogate Assisted Generation of Human-Robot Interaction Scenarios"
_robot-learning.org/CoRL/2023/Conference — CoRL 2023 Oral_

### Official Review · Reviewer_EBp2 · 2023-07-11

**Confidence:** 4
**Originality:** Good
**Technical Quality:** Good
**Clarity Of Presentation:** Good
**Impact:** 4

**Recommendation:**

Strong Accept: I recommend accepting the paper and will argue for my recommendation even if other reviewers hold a different opinion.

**Review:**

I highlighted my comments below:

Major:
From the perspective of shared control and human-robot interaction (HRI), this work makes a valuable contribution to the field by addressing the creation of challenging datasets for meaningful algorithm testing and evaluation. While previous work primarily focused on shared autonomy (SA), extending it to HRI is a reasonable direction. However, I am curious about how the proposed scenario generation approach would adapt to more complex and difficult tasks. As mentioned by the authors, improving realism would require incorporating more complex environments, human actions, and sophisticated simulators. Nonetheless, for the current work, the low-dimensional workspace was discretized. Considering that more complex tasks could lead to the curse of dimensionality, it may potentially impact performance. Could you please elaborate further on this aspect?

Furthermore, the formulation of a quality diversity (QD) problem seems reasonable and yields notable results. However, in the current work, the optimization is performed over two user-defined measures, leading to convergence after a considerable amount of time (>12 hours). I am curious about the scalability of the proposed method in relation to the dimensionality of the given measures. Specifically, I would like to understand whether the proposed methods exhibit similar sample inefficiency as seen in Domain Randomization when faced with high-dimensional search spaces. Exploring how the proposed methods perform and handle increased dimensionality would provide valuable insights into their efficacy and potential limitations considering more complex HRI tasks. I kindly request the authors to address these questions and provide further clarification on the scalability of the proposed methods with respect to measure dimensionality.

Both proposed methods, DSAS and SAS, exhibit superior performance compared to the baselines in terms of sample efficiency and wall-clock time. However, when compared to prior work [1, 2], the introduction of the surrogate model increases the computational demand. As stated in line 198, the proposed methods were tested five times due to GPU usage constraints. It would be helpful if the authors could provide more insights into these limitations and discuss whether they could pose challenges in scenario generation for more complex HRI settings.

Minor:
The paper is well-written, and the visualizations are well-made. However, I would like to suggest that the authors consider adjusting the placement of the figures to avoid having two figures on the same page below each other. This adjustment could enhance the readability and overall quality of the paper. Additionally, it would be beneficial to revise the scale of Figure 5, as the images appear quite small and are challenging to perceive. By making these adjustments, the figures will be more accessible and contribute to a better understanding of the paper's content.

References:

[1] M. C. Fontaine and S. Nikolaidis. A quality diversity approach to automatically generating human-robot interaction scenarios in shared autonomy. In Robotics: Science and Systems, 2021

[2] M. C. Fontaine and S. Nikolaidis. Evaluating human–robot interaction algorithms in shared autonomy via quality diversity scenario generation. ACM Transactions on Human-Robot Interaction (THRI), 11(3):1–30, 2022.

[3] V. Bhatt, B. Tjanaka, M. C. Fontaine, and S. Nikolaidis. Deep surrogate assisted generation of environments. Conference on Neural Information Processing (NeurIPS), 2022

**Quality Of The Limitations Section:**

Additional details required

**Questions For Rebuttal:**

I summarize my major questions below:
1. I am curious about the scalability of the proposed algorithms,either in terms of task complexity or the number of measures.
2. It would be greatly appreciated if you could provide additional insights into the computational demand of the proposed methods. Considering the potential challenges for more complex tasks, how do you foresee this computational demand becoming problematic?

**Robotics Focus:**

Sufficient demonstration on hardware

**Summary Of Paper:**

This work addresses the challenge of generating datasets containing challenging scenarios for shared control and human-robot interaction. To tackle this issue, the authors propose two variations of surrogate-assisted scenario generation algorithms that formalize model-based quality diversity (QD) problems. Unlike previous approaches that either do not leverage a surrogate model [1,2] or do not consider continuous state and action spaces [3], the proposed methods combine these elements effectively. The proposed approach follows a hierarchical process, with the inner loop utilizing the QD algorithm to identify diverse and challenging scenarios based on the surrogate model, while the outer loop evaluates and adds these scenarios to the dataset used for surrogate model training. The experimental results demonstrate that both methods outperform baselines in terms of sample efficiency and wall-clock time.

**Summary Of Recommendation:**

The paper is well-written and structured. Aside from some minor comments regarding the figures, my main concern revolves around the limitations of the algorithms in terms of scalability and computational demand. That being said, I recognize the value of the paper's contribution to the field of human-robot interaction (HRI) in terms of creating valuable scenarios for evaluating and benchmarking proposed methods. Therefore, I'm leaning towards accepting the paper, albeit weakly. I'm open to changing my decision to a full acceptance if the authors provide a compelling rebuttal that effectively addresses the concerns raised.

---

> ### Author Response · Authors · 2023-08-08
> **Response to Reviewer EBp2 (1/2)**
>
> > Considering that more complex tasks could lead to the curse of dimensionality, it may potentially impact performance. Could you please elaborate further on this aspect?
>
> We thank the reviewer for the great feedback. We hypothesize that as high-fidelity HRI simulators become available and we can simulate complex domains, the advantages of our approach using surrogate models will be significantly enhanced for the following reasons: First, scenarios of more complex domains will likely have more scenario parameters. Contrary to the baselines, our algorithm, DSAS, scales better to higher dimensional scenario parameter spaces since we apply DQD [4] while searching for scenarios in the inner loop. DQD has been shown to achieve state-of-the-art performance in search parameter spaces with ~10,000 parameters [41]. The reason is that in DQD, we use objective and measure gradients directly. Thus, our search is applied over the objective-measure space (k + 1 dimensions in our problem formulation) instead of the scenario parameter space (n dimensions in our problem formulation). Second, more complex domains require longer simulation times. Thus, we anticipate that the improvements we have seen with respect to the wall-clock time in the shared workspace collaboration domain because of the surrogate model predictions will be even more pronounced in these domains.
>
> > I am curious about the scalability of the proposed method in relation to the dimensionality of the given measures
>
> In our domains, we were able to specify good test coverage with low-dimensional measure spaces. Thus, we followed standard QD methods that employ grid archives with uniformly-spaced cells. We would like to clarify that the measures represent attributes that constitute test coverage in a given domain, and a more complex domain that could have a high-dimensional scenario parameter space may not necessarily require a higher-dimensional measure space.
>
> However, domains where we wish to obtain good test coverage over a large number of attributes would indeed require high-dimensional measure spaces. When using grid archives, increasing the number of measures results in an exponential increase in the number of cells in the archive, as the reviewer observes. Attempting to cover a high-dimensional grid archive with SAS and DSAS, as well as with any of the baselines, would require a very large number of evaluations to fill in that space. In such cases, we can leverage the following advances by the QD community to extend our algorithms to high-dimensional measure spaces.
>
> Specifically, we can observe that the curse of dimensionality arises from the fact that the measure space is partitioned into uniformly-spaced grid cells with a high resolution. We believe that in many tasks, this assumption is unnecessary. Some parts of the measure space may be irrelevant, or they may allow for a much lower resolution. E.g., in our task, we may not need centimeter accuracy in the distance between objects. Rather than using a standard archive with uniformly spaced grid cells, we can use a centroidal Voronoi tessellation (CVT) to partition the measure space into a predefined number of regions. Previous work (“Using Centroidal Voronoi Tessellations to Scale Up the Multidimensional Archive of Phenotypic Elites Algorithm” by Vassiliades et al.) has shown that CVT-based QD algorithms can scale to measure spaces of hundreds of dimensions. We can trivially extend SAS and DSAS to interact with a CVT-based archive.
>
> Alternatively, if the intrinsic dimensionality of the measure space is low, then the measure space can be represented by a low dimensional manifold. Dimensionality reduction methods for QD algorithms like AURORA (“Autonomous skill discovery with Quality-Diversity and Unsupervised Descriptors” by Cully) can reduce the dimensionality of the measure space before applying QD.
>
> Overall, dealing with high-dimensional measure spaces is an active area of research in QD optimization, and any improvements in this area can be applied to future QD scenario generation systems.

---

> ### Author Response · Authors · 2023-08-08
> **Response to Reviewer EBp2 (2/2)**
>
> > It would be greatly appreciated if you could provide additional insights into the computational demand of the proposed methods. Considering the potential challenges for more complex tasks, how do you foresee this computational demand becoming problematic?
>
> SAS and DSAS leverage a deep neural network based surrogate model. They require a GPU for efficient training of the model. However, due to better sample efficiency, SAS and DSAS require fewer scenario evaluations to obtain results similar to the baselines. In domains where evaluations are cheap, e.g., shared teleoperation, the compute and the time required to train and leverage a surrogate model outweigh the savings due to higher sample efficiency, making DSAS and SAS not well suited for such domains. However, in more complex domains, scenario evaluations require a lot more computation time. Since SAS and DSAS are more sample efficient, we can reduce the number of scenario evaluations and still obtain similar results as the baselines. The savings from reducing the number of evaluations more than makes up for the extra time needed to train the surrogate model and run the inner loop. We observe this in shared workspace collaboration, which is still relatively simple compared to real-life HRI tasks. We will clarify the computational requirements of SAS and DSAS in Sec. 6 of the revised version of the paper.
>
> We note that the line about GPU constraints (line 198) was a result of our access to computational resources. We ran our experiments on a shared high-performance server in which machines with GPUs are always in high demand. Hence, it was easier to schedule more runs that only required CPUs, regardless of the total time required for the experiment.
>
> We would like to emphasize that the surrogate model visualized in Fig. 6, Appendix B, is currently small enough to be run on most desktop GPUs. However, we recognize that tackling more complex tasks may require larger neural network surrogate models. At the same time, we anticipate that for such tasks, GPUs would be necessary for efficient evaluation in the HRI simulator itself. Thus, the increased computational demand would apply to any algorithm utilizing the simulator for testing.  As the capability to simulate more complex domains expands through the emergence of high-fidelity simulators, we intend to further investigate the comparison between the compute requirements for simulation and the computational demands involved in training and retaining a surrogate model for prediction.
>
> We will additionally make all figure adjustments suggested by the reviewer.

---

### Official Review · Reviewer_rtcn · 2023-07-18

**Confidence:** 2
**Originality:** Fair
**Technical Quality:** Good
**Clarity Of Presentation:** Good
**Impact:** 2

**Recommendation:**

Weak Accept: I recommend accepting the paper, but will not argue for my recommendation if the majority of other reviewers have a different opinion.

**Review:**

HRI scenarios are highly relevant. The manuscript is clearly written and the work is properly motivated. The experiments show that the proposed methods provide more diverse scenarios compared to the chosen baseline methods. A small subset of the scenarios found by the proposed approach is evaluated on a real setup.

The concrete collaborative tasks in section 6 are not properly described. Without looking at Figure 5 and the accompanied video, it stays unclear what the actual task is. Even with these visual guides, it is unclear what an optimal solution to the HRI task would look like.

**Quality Of The Limitations Section:**

Limitations are addressed clearly

**Questions For Rebuttal:**

- In Figure 3, I assume that iterations and wall time should somehow correlate, meaning that when comparing the top and bottom rows, only the x-axis (iterations or time) should scale. This is true for Figures 3b and 3c. However, in Figure 3a DSAS has the highest QD score for most of the iterations (top row) while it has the lowest QD score over time (bottom row). Why is that?

- The supplement video shows four "successful" failure scenarios that have been sampled from the predicted scenario parameter space. What would non-failure scenarios look like?

- The authors claim that the proposed approach is computationally more efficient. This is shown by the "quality diversity" score over time in Figure 3, which shows a higher score at nearly any point in time for the proposed approach. However, the evaluation in the real scenario only shows that the sampled scenarios from the proposed approach are reproducible in the real world. It does not show if scenarios sampled from the baseline approaches are also reproducible in the real world or different from the proposed approach. Considering e.g. Figure 3c bottom row, how would SAS and MAP-Elites compare after 3h when reproduced in the real world? SAS here clearly has a higher QD score but it is unclear how this impacts a scenario in the real world.

- Joint limits and self-collisions are trivial problems that can be avoided by standard motion planners. Why can this not be incorporated into the HRI task?

**Robotics Focus:**

Sufficient demonstration on hardware

**Summary Of Paper:**

The paper's premise is that the evaluation of human-robot interactive systems (HRI) is difficult and costly. Therefore, the paper proposes a surrogate model to predict the behaviour of the human and the robot. The overall goal is to find diverse scenarios for training. The work is evaluated on two scenarios using a Kinova JACO robot arm.

**Summary Of Recommendation:**

While the graphs of the QD score over time show an improvement for every point in time for the proposed approach and therefore would support the author's claims, the real-world impact of this score is not clear to me. However, I am not an expert in this area and will adjust my verdict given the recommendations of reviewers more confident in this area.

---

> ### Author Response · Authors · 2023-08-08
> **Response to Reviewer rtcn (1/2)**
>
> > in Figure 3a DSAS has the highest QD score for most of the iterations (top row) while it has the lowest QD score over time (bottom row). Why is that?
>
> The x-axis in the top row is the number of ground-truth scenario evaluations in the simulator. The baselines evaluate every proposed scenario in the simulator. Our proposed algorithms, SAS and DSAS, only do this in the outer loop. In the inner loop, they exploit the surrogate model with many iterations of a QD algorithm, where scenario evaluations are predicted via a forward pass of a neural network. The inner loop and the model retraining in SAS and DSAS take a non-negligible amount of time but perform no ground-truth scenario evaluations. Thus, for a fixed number of evaluations, DSAS and SAS require more wall-clock time compared to other algorithms, due to the overhead of model evaluations and model training.
>
> There is an inherent tradeoff in model-based methods between computational time to retain a model and sample efficiency, when compared to their model-free counterparts. Training and searching with a surrogate model increases algorithmic overhead but it requires fewer simulated evaluations, which are expensive in domains with long scenarios.
>
> This is illustrated in our domains: In the shared teleoperation domain (Figure 3a), evaluations in the simulator last only a few seconds. Hence, the additional overhead of the surrogate model results in DSAS and SAS performing worse with respect to the wall-clock time. On the other hand, in the shared workspace collaboration domain (Fig. 3b and 3c), each scenario simulation requires several minutes. Hence, the improved sample efficiency – in terms of simulated scenarios – of SAS and DSAS outweighs the extra time needed for the inner loop and retraining.
>
> We will clarify this result in the revised version and thank the reviewer for pointing this out.
>
> > The concrete collaborative tasks in section 6 are not properly described. Without looking at Figure 5 and the accompanied video, it stays unclear what the actual task is. Even with these visual guides, it is unclear what an optimal solution to the HRI task would look like.
>
> We thank the reviewer for the feedback. We would like to point out that we adopt the tasks from prior work in shared control teleoperation [51] and workspace collaboration [42]. We provide a short description in section 5 and a more detailed description in Appendix D. We will add pictures of the task in Appendix D.
>
> In both tasks, the goal of the HRI algorithm is to minimize task completion time:
>
> The shared teleoperation domain requires the robot to accurately infer the user’s intended goal and reach that goal. An optimal robot correctly infers the user’s goal and follows the optimal path to that goal.
>
> The shared workspace collaboration domain requires the robot to accurately infer the user’s intended goal and reach one of the alternative goals. The optimal HRI solution depends on the goal object configuration and the user’s behavior since the user here also affects the state of the world. An optimal robot moves concurrently with the user to an alternative goal, following the optimal path to that goal.
>
> We provide examples of two successful task executions in the shared workspace collaboration domain in Appendix I. We also attach the videos of the two tasks reproduced in the real world in _rebuttal.zip_ (in a separate response).
>
> > The supplement video shows four "successful" failure scenarios that have been sampled from the predicted scenario parameter space. What would non-failure scenarios look like?
>
> In App. I, Fig. 10, we show examples of scenarios in which the HRI algorithm performs well. In
> the first scenario (Fig. 10a), we observe that the task completion time is low if the goal objects are spaced apart since this allows the robot to accurately infer the user’s goal. In the second scenario (Fig. 10b), we observe low task completion time (high performance) when the goal objects are very close. Inferring the user’s goal is difficult in this scenario, but mistakes in inference can be quickly corrected since other goals are close. We attach the recorded videos of these scenarios in the real world in _rebuttal.zip_ (in a separate response) for clarity purposes.

---

> ### Author Response · Authors · 2023-08-08
> **Response to Reviewer rtcn (2/2)**
>
> > ... This is shown by the "quality diversity" score over time in Figure 3, which shows a higher score at nearly any point in time for the proposed approach. However, the evaluation in the real scenario only shows that the sampled scenarios from the proposed approach are reproducible in the real world. It does not show if scenarios sampled from the baseline approaches are also reproducible in the real world or different from the proposed approach ... has a higher QD score but it is unclear how this impacts a scenario in the real world.
>
> We first provide intuition on the quality diversity (QD) score. QD-score is defined as the sum of the objective values of all occupied cells in the archive. A higher QD score can be obtained by increasing the number of occupied cells or improving the objective value of existing cells. In scenario generation, this means either newly discovered scenarios or worse performance of the tested system for existing measures.
>
> We observe a visualization of this concept in the archive heatmaps in Fig. 4. Each cell is a scenario, and the color of a cell indicates the time taken to complete the scenario in that cell. Yellow color implies that the HRI algorithm failed to complete the scenario within the time limit, whereas no color implies that the QD algorithm did not find a scenario with those measure values. We observe that the SAS and DSAS archives contain scenarios in cells that are unoccupied in the corresponding cells of the baseline archives. This means that scenarios with the same measure values do not exist in the baseline archives. For example, in the SAS and DSAS heatmaps of Fig. 4, there are yellow cells close to the bottom-right area of the heatmap, where human variation (noise in human inputs) in the y-axis is nearly zero and the distance between goals in the x-axis is large. This is a failure where the user is nearly-optimal and the objects are far apart. We observe that in MAP-Elites, there are no yellow cells in that area. This implies that no failure was found by MAP-Elites for the case where the user is nearly-optimal and the objects are far apart.
>
> Beyond individual failures, a filled archive allows a designer to draw general, qualitative conclusions about the system’s behavior. As discussed above, the SAS and DSAS archives have scenarios that correspond to nearly-optimal human actions. If a designer observed the archive generated by Random Search, they would not have any information about the algorithm’s performance for a nearly optimal user.
>
> Another example from Fig. 4 is the effect of distance between goal objects. If a system designer observed the archive from MAP-Elites, they may erroneously believe that placing objects more than 0.16m away would ensure the HRI algorithm would always complete the task within the specified time limit since there are no yellow cells after that distance. However, the DSAS and SAS archives are filled with yellow cells even when the objects are placed up to 0.25m apart.
>
> We will clarify the connection between QD-score, heatmaps, and failures in the revised version and we thank the reviewer for bringing up this important point.
>
> > Joint limits and self-collisions are trivial problems that can be avoided by standard motion planners. Why can this not be incorporated into the HRI task?
>
> We would like to point out that we focus on testing previously proposed HRI algorithms, which we view as black boxes. Specifically, we test popular algorithms [42, 51, 52, 53] (see App. E.1 for details about the robot’s policy) that are designed to adapt to the user’s goal online by either predicting the user’s goal and blending a robot control command with the user’s input [52], or inferring a distribution of user goals and solving for a robot control command that minimizes the total expected cost-to-go [42,51,53]. Previous work has shown the latter approach to perform robustly in shared teleoperation and shared workspace manipulation tasks.
>
> While these algorithms do not explicitly reason about joint limits or self-collision when computing the robot control command, we used a standard motion planner to re-plan around self-collisions or joint limits when these were detected. These replanning steps induce a delay in task completion. As the reviewer suggests, there are alternative approaches for addressing joint limits and self-collisions. For instance, we could combine goal inference with sampling-based costmap planners that generate collision-free paths. This approach would avoid the delay from replanning when a self-collision or joint limit is detected, but it would require additional time to compute the collision-free motion plans.
>
> In this work, we take existing implementations of prior systems and focus on algorithmic scenario generation to discover the limitations of these implementations. Once a designer is aware of the various system limitations revealed by the generated scenarios, they can make adjustments to improve system performance.

---

### Official Review · Reviewer_ZDjU · 2023-07-20

**Confidence:** 3
**Originality:** Good
**Technical Quality:** Good
**Clarity Of Presentation:** Good
**Impact:** 3

**Recommendation:**

Weak Accept: I recommend accepting the paper, but will not argue for my recommendation if the majority of other reviewers have a different opinion.

**Review:**

Strengths:
- In general, I found the paper to be well written and easy to follow.
- The proposed approach seems like an elegant combination of the ideas in [1] and [21] with sensible advances to support application to more complex HRI systems
- The empirical results presented in the paper are promising

Scope for improvement:
- The approach is only evaluated on two (relatively simple) domains. I would like to see the approach evaluated on a larger and more diverse set of domains to better support the claims that the algorithms generally apply to HRI systems.
- Because the paper is motivated by HRI systems, I would like to see some empirical evidence that the distribution of synthesized challenging human actions is similar to the distribution of challenging human actions that you would actually encounter “in the wild”. For the real world demo reported in the paper, a user is asked to reproduce scenarios extracted from the generated archives. This shows that the generated scenarios are reproducible, but it does not indicate that the generated scenarios cover situations that you would naturally encounter. I think the case for the proposed approach would be much stronger if the authors included some larger scale user study that shows that the types of difficult scenarios encountered naturally are actually covered by the distribution of synthesized scenarios.
- The paper references an appendix multiple times (e.g. “See App. A for the complete pseudocode”), but there is no appendix included in the submitted document. After downloading the supplementary material I see the appendix is included in the zip file. The other submissions I am reviewing have the appendix included in the document (which makes the reference links work). Please include the appendix in the submitted document.
- I personally find the diagram in Figure 2 to be difficult to follow. You might consider breaking it up into multiple diagrams.

**Quality Of The Limitations Section:**

Limitations are addressed clearly

**Questions For Rebuttal:**

I would ideally like to see a more rigorous evaluation included in the paper, but I think that is probably out of scope for the rebuttal period.  In lieu of that, you might consider adding some discussion to make it more clear how you expect the approach to generalize to other HRI domains (e.g. describe some example measures m you might use for a more diverse set of domains).

I like that you include an ablation study to help understand how the objective regularization contributes to the results of your experiments. Would it make sense to also include ablations without scenario repair to better understand the effect of that part of your approach?

**Robotics Focus:**

Sufficient demonstration on hardware

**Summary Of Paper:**

This paper considers the problem of generating diverse/challenging scenarios for evaluating HRI tasks.  Previous works formulated the problem as a quality diversity (QD) problem which has been shown effective in a simple teleop HRI domain. However, previous methods require directly evaluating robot policies and human actions, which is expensive and limits the scalability to more complex tasks. The authors propose to instead train a model to predict task outcomes and use that model in place of expensive evaluations.

The authors evaluate their approach in two domains: a shared control teleoperation task and a shared workspace collaboration task. The reported experiments show the proposed approaches outperforming baseline QD algorithms in sample efficiency. Additionally, wall clock time improvement is shown on the collaboration task where evaluation is particularly expensive.

**Summary Of Recommendation:**

The problem is well motivated and relevant. I think the proposed approach is in some ways an incremental work combining the ideas of [1] and [21]. However, the authors propose sensible advances to adapt the ideas for application to more complex HRI systems and the empirical results presented in the paper are promising. For these reasons, I think the paper is interesting and worth sharing with the community.

---

> ### Author Response · Authors · 2023-08-08
> **Response to Reviewer ZDjU (1/2)**
>
> > The approach is only evaluated on two (relatively simple) domains. I would like to see the approach evaluated on a larger and more diverse set of domains to better support the claims that the algorithms generally apply to HRI systems.
>
> We would like to highlight that our DSAS algorithm treats the underlying HRI Domain, including the algorithm being evaluated, as a black-box and only assumes access to the objective and measure functions from the QD definition as well as the executed human and robot trajectories after the scenario has been evaluated. We selected the shared teleoperation and shared workspace collaboration domains because we believe that they are representative of a wide range of tasks at home and in the workspace and they match popular domains from prior work in the HRI literature. We limit the number of domains to show discovered failures in each setting while still fitting within the space constraints of a conference paper’s scope.
>
> Testing our algorithm on more complex domains is limited by our ability to simulate these domains – as the complexity of HRI domains increases, so does the need for more sophisticated human models that simulate realistic human behaviors. Simulating human behavior is a challenging problem and a rapidly developing field of research in HRI, which we consider orthogonal to this work. We believe advances in human modeling in HRI will allow us to scale future scenario generation systems to more complex scenarios, as our system treats the human model as a black-box and searches over parameterized human models.
>
> > You might consider adding some discussion to make it more clear how you expect the approach to generalize to other HRI domains (e.g., describe some example measures m you might use for a more diverse set of domains).
>
> We thank the reviewer for the great suggestion. We discuss example QD formulations for two alternative domains. The measures capture aspects of the environment, human and robot behaviors, and represent factors that we expect to affect the team performance:
>
> _Robot navigation around pedestrians_ (Kruse et al., Gao et al.): The objective here would be to minimize the time required for the robot to reach its goal while avoiding collisions with pedestrians. Measures could include the number and size of obstacles in the scene, the robot’s goal location, the minimum distance between the mobile robot and the pedestrians during navigation, the average speed of pedestrians, the curvature of the pedestrian trajectories, the total number of seconds the robot remained idle during the task, etc.
>
> _Assistive Feeding_ (Gallenberger et al.): The objective here would be to successfully transfer all bites on a plate to the user. Measures could include aspects of the user’s capability (e.g., how far the user can reach to receive a bite, how long the user takes to receive a bite), physical user characteristics such as their height, time spent by the user being idle waiting for a bite, etc.
>
> We will include this discussion in the revised version. We are excited about future work exploring scenario generation in these domains.
>
> - Kruse, T., Pandey, A. K., Alami, R., & Kirsch, A. (2013). Human-aware robot navigation: A survey. Robotics and Autonomous Systems.
> - Gao, Y., & Huang, C. M. (2022). Evaluation of socially-aware robot navigation. Frontiers in Robotics and AI.
> - Gallenberger, D., Bhattacharjee, T., Kim, Y., & Srinivasa, S. S. (2019). Transfer depends on acquisition: Analyzing manipulation strategies for robotic feeding. In Proceedings of the 14th ACM/IEEE International Conference on Human-Robot Interaction (HRI).
>
> > I think the case for the proposed approach would be much stronger if the authors included some larger scale user study that shows that the types of difficult scenarios encountered naturally are actually covered by the distribution of synthesized scenarios.
>
> We agree with the reviewer that a large-scale user study that shows that the synthesized scenarios naturally occur with real users would be beneficial. We argue that the design and implementation of such a study would be a separate contribution. The reason is that the algorithms we test have been shown to work robustly in practice [42, 51, 52, 53], and the proposed approach discovers edge-case failures that are hard to find and relatively rare in practice. Thus, a user study where users can freely interact with the system would need to involve a very large number of subjects to observe these failures. Alternatively, we could constrain the study so that subjects are more likely to exhibit these behaviors; on the extreme end of this spectrum is reproducing these failures, as we do in this paper. Furthermore, we would need to account for any safety concerns arising from the unexpected robot behavior in the corner cases. We view solving these challenges as beyond the scope of this paper, and here we show that these failures will actually occur in the real world if users act in a certain way.

---

> ### Author Response · Authors · 2023-08-08
> **Response to Reviewer ZDjU (2/2)**
>
> > Would it make sense to also include ablations without scenario repair to better understand the effect of that part of your approach?
>
> Our approach assumes that the generated scenarios are valid since we are unable to evaluate invalid scenarios. Without the scenario repair, we would end with scenarios that are impossible, e.g., overlapping goal objects, or trivially hard, e.g., goal objects outside the workspace boundaries that are unreachable by the robot. The scenario repair guarantees that all generated scenarios are valid and thus can be passed to the simulator. We will clarify this in the revised version, and we thank the reviewer for the great comment.
>
> > Please include the appendix in the submitted document.
>
> Thank you very much for the suggestion. We were unaware that we could include the appendix in the main submitted document; we will do so in the revised version.
>
> > I personally find the diagram in Figure 2 to be difficult to follow. You might consider breaking it up into multiple diagrams.
>
> Thank you for the feedback. We will separate the inner loop and the outer loop into two diagrams in Appendix A and provide a more detailed explanation.

---

### Official Review · Reviewer_3aQ9 · 2023-07-20

**Confidence:** 3
**Originality:** Excellent
**Technical Quality:** Excellent
**Clarity Of Presentation:** Excellent
**Impact:** 4

**Recommendation:**

Strong Accept: I recommend accepting the paper and will argue for my recommendation even if other reviewers hold a different opinion.

**Review:**

**Strengths**

* The paper provides a novel and promising method to "Oz the Wizard" (Steinfeld, Aaron, Odest Chadwicke Jenkins, and Brian Scassellati. "The oz of wizard: simulating the human for interaction research." Proceedings of the 4th ACM/IEEE international conference on Human robot interaction. 2009.)
* The paper builds on prior work in the field and yet seems to be making a significant contribution
* The paper surveys the literature on this topic adequately
* The paper's evaluations are appropriate and the results are great
* The paper characterizes its limitations appropriately


**Quality Of The Limitations Section:**

Limitations are addressed clearly

**Questions For Rebuttal:**

None

**Robotics Focus:**

Sufficient demonstration on hardware

**Summary Of Paper:**

* The paper provides an extension on prior work with the cotribution of a new Quality diversity environment generation algorithm that receives feedback via surrogate models for a human and a robot that are trained on the generated environments
* In particular, providing a quality diversity algorithm with (a) feedback on the feasibility of the generated scenario, via corrections, and (b) the "true" quality of the scenario via simulation, the authors show that scenario generation quality is better, the algorithm is more sample efficient, and for complex domains, the algorithm is time efficient too.

**Summary Of Recommendation:**

I strongly recommend that this paper be included in the technical program.

---

> ### Author Response · Authors · 2023-08-08
> **Response to Reviewer 3aQ9**
>
> We thank the reviewer for the very encouraging feedback. We will highlight the connection with the “Oz of the Wizard” paper that proposes using potentially simplified models of human behavior to test variability and feasibility in HRI systems.

---

> > ### Comment · Reviewer_3aQ9 · 2023-08-13
> > **No change in rating**
> >
> > I have read the reviews of my fellow reviewers as well as the authors' comments in response to them. I am not changing my rating as I  think this paper is still a very valuable contribution to the field.

---

### Author Response · Authors · 2023-08-08
**Thank you very much for the detailed reviews.**

Thank you very much for the thorough reviews and insightful comments. We are excited that reviewers found that our work “makes a valuable contribution to the field”, it provides “an elegant combination of the ideas in [1] and [21] with sensible advances” and that the “manuscript is clearly written.” The reviewers raise some great questions and we respond to each reviewer separately below. We look forward to continuing the discussion.

---

### Decision · Program_Chairs · 2023-08-30

**Decision:**

Accept (Oral)

**Comment:**

Summary of Paper:
The paper delves into generating diverse and challenging scenarios for evaluating Human-Robot Interaction (HRI) tasks. The authors extend previous work by introducing a new Quality Diversity environment generation algorithm, taking feedback from surrogate models trained on the generated environments for humans and robots. Their approach bypasses the expensive evaluations of direct robot policies and human actions that hampered the scalability of prior methods. Through evaluations in two domains, shared control teleoperation and shared workspace collaboration, the paper showcases enhanced sample efficiency and reduced wall clock time compared to baseline Quality Diversity algorithms.

Strengths:
- The proposed method offers an inventive approach to generating challenging datasets for HRI, rooted in the Quality Diversity algorithm, which receives feedback via surrogate models. This is particularly notable considering its grounding in the "Oz the Wizard" framework.
- An impressive build on existing research, the paper's contribution is significant, aptly drawing from and expanding upon the foundational literature.
- The paper is well organized, ensuring comprehension and presenting a clear narrative of the proposed method's efficacy, evidenced by empirical results that underscore its merits over QD algorithms in terms of sample efficiency and timing.
- Authors are transparent about the limitations of their work, which underscores the paper's credibility.

Areas for Improvement:
- Domain Limitations: The proposed approach, while promising, has been tested only on two domains. Expanding evaluation across more diverse and complex domains would bolster claims about the algorithm's broad applicability in HRI systems.
- Empirical Grounding in Real-World Scenarios: There's a palpable gap between the generated scenarios' reproducibility and their alignment with real-world HRI challenges. A more comprehensive user study would validate whether the synthesized scenarios encapsulate the range of challenges one would face in actual environments.
- Clarity and Documentation: Several aspects, such as a clearer description of concrete collaborative tasks and the inclusion of referenced appendices directly within the paper, could enhance the submission's clarity and completeness. Some visual elements, like Figure 2, could be reworked for better comprehension.
- Computational Concerns: While the paper boasts of improved sample efficiency and reduced time, there are implicit concerns about the computational demands, especially when juxtaposed with prior works. Addressing the scalability and computational constraints, especially in more intricate HRI contexts, would be beneficial.
- Dimensionality and Complexity: The current work's limited scope, in terms of its low-dimensional workspace, raises questions about the algorithm's performance as the complexity and dimensionality of tasks increase. Engaging with these concerns might provide a more comprehensive view of the approach's versatility.

Discussion during the rebuttal phase:
The authors responded effectively to the queries raised by the reviewers, resulting in a significant improvement in the reviewers' rating scores. The authors have indicated in the rebuttal phase that they will add further references to related works and provide more detailed data in the appendix for the camera-ready manuscript. Consequently, we can expect an even higher-quality paper in the end.

Recommendation:
The paper showcases a commendable effort in advancing HRI scenario generation. Given the paper's considerable strengths, especially its novel approach and strong empirical grounding, coupled with its addressable weaknesses, I recommend accepting the paper. However, it would benefit the authors to consider the discussion with reviewers, addressing key concerns to ensure the work's robustness and broader applicability.